# Neural Thermodynamics: Entropic Forces in Deep and Universal Representation Learning

Liu Ziyin[1,3,*], Yizhou Xu[2,*], Isaac Chuang[1]

[1]*Massachusetts Institute of Technology*
[2]*École Polytechnique Fédérale de Lausanne*
[3]*NTT Research*

## Abstract

With the rapid discovery of emergent phenomena in deep learning and large language models, understanding their cause has become an urgent need. Here, we propose a rigorous entropic-force theory for understanding the learning dynamics of neural networks trained with stochastic gradient descent (SGD) and its variants. Building on the theory of parameter symmetries and an entropic loss landscape, we show that representation learning is crucially governed by emergent entropic forces arising from stochasticity and discrete-time updates. These forces systematically break continuous parameter symmetries and preserve discrete ones, leading to a series of gradient balance phenomena that resemble the equipartition property of thermal systems. These phenomena, in turn, (a) explain the universal alignment of neural representations between AI models and lead to a proof of the Platonic Representation Hypothesis, and (b) reconcile the seemingly contradictory observations of sharpness- and flatness-seeking behavior of deep learning optimization. Our theory and experiments demonstrate that a combination of entropic forces and symmetry breaking is key to understanding emergent phenomena in deep learning.

## 1   Introduction

Modern neural networks trained with stochastic gradient descent (SGD) exhibit a complex plethora of emergent behaviors – emergence of capabilities [1, 2, 3], progressive sharpening and flattening [4, 5], phase-transition like behaviors [6, 7], and universal representational alignment across models [8, 9, 10, 11] – that are difficult to explain through loss minimization alone. These behaviors mirror phenomena found in physical systems at finite temperature, suggesting that deep learning dynamics are shaped not just by explicit optimization but also by implicit forces arising from stochasticity and discrete updates. These implicit forces have long been associated with the phenomenon of "implicit bias" in deep learning [12, 13, 14, 15], but their precise mathematical nature remains elusive. In physics, such effects are often captured by *entropic forces*—macroscopic forces that emerge from the system's statistical tendencies rather than its energy landscape alone [16]. The power of this framework lies in the notion of an *effective entropy*, which plays the role of a potential whose gradients define the entropic force. Identifying this effective entropy not only reveals what the system is implicitly optimizing, but also opens the door to leveraging theoretical tools from statistical physics to analyze and improve AI models.

**Contributions.**   We formalize this connection between stochastic learning dynamics and entropic forces through the lens of symmetry and representation learning to:

39th Conference on Neural Information Processing Systems (NeurIPS 2025).

---

*Equal contribution.

1. Derive an entropic loss function and show that the entropic forces of SGD *break continuous parameter symmetries while preserving discrete ones* (Section 3).
2. Show that the symmetry breaking due to entropic forces gives rise to a family of *equipartition theorems* that predict the gradient alignment phenomena (Section 4).
3. Explain and unify two seemingly disparate but universal observations – *progressive sharpening of the loss landscape* and the *emergence of universal representations* – as consequences of entropic forces (Section 5).

Our theory establishes a principled framework – akin to a thermodynamics of deep learning – that unifies several universal phenomena under a single formalism. The results suggest that the entropic loss landscape, shaped by both optimization and entropy, plays a foundational role in understanding learning dynamics and emergent phenomena. Full derivations and experimental validations are provided in the appendix.

## 2 Related Work

**Modified Loss and Effective Landscape.** The concept of modified or effective losses has emerged as a critical framework for understanding the implicit biases induced by stochastic gradient descent (SGD) in deep learning, which differs from another line of work [17, 18, 19, 20, 21] which leverages the property of stationarity to analyze the stationary distribution of SGD. Ref. [22] introduced the notion of a modified loss to analyze the discrete-time dynamics of SGD, demonstrating how discretization implicitly alters the optimization landscape. Similarly, Refs. [23] and [24] extended the modified loss formulation to where there is a gradient noise due to minibatch sampling. These works conducted numerical simulations to show that training on the effective loss really approximates the original dynamics [23, 25, 26] and leads to similar generalization performances. In this work, we refer to this type of losses as entropic losses for their associations with theoretical physics. Still, these entropic losses remain poorly understood, and their significance for understanding emergent phenomena in deep learning is not yet appreciated. Our work finds the crucial link between the entropic loss and symmetry-breaking dynamics, which is important for understanding the various intriguing nonlinear phenomena of representation learning.

**Parameter Symmetry in Neural Networks.** Parameter symmetries are shown to play a fundamental role in shaping neural network training dynamics and their emergent properties [27, 28, 29, 7, 30, 31]. A series of works showed that continuous symmetries in the loss function give rise to conservation laws, which imply that the learning result of SGD training is strongly initialization-dependent [32, 28, 33]. More recent works showed how any stochasticity or discretization effect could break the symmetries in a systematic way such that the learned solution is no longer dependent on the initialization, a hint of universality [34, 29, 7, 31]. Particularly, Ref. [29] developed the formalism of exponential symmetries and proved that any loss function with an exponential symmetry leads to a symmetry-breaking dynamics that converges to unique fixed points. This point can be seen as the dynamical equivalence of our Theorem 2, which states that there is essentially no continuous symmetry in the entropic loss. In comparison, our framework takes a different perspective: we study the symmetry from a loss landscape perspective and identify these symmetry-breaking tendencies as entropic forces. This unified perspective enables us to understand the universal learning phenomena with a greatly simplified analysis.

## 3 Effective Energy for Stochastic Gradient Learning

Define $\ell(x, \theta)$ to be the per-sample loss function. We can define the empirical risk as

$$L(\theta) = \mathbb{E}_{\mathcal{B}}[\mathbb{E}_{x \in \mathcal{B}} \ell_\gamma(x, \theta)], \tag{1}$$

where $\ell_\gamma(x, \theta) \coloneqq \ell(x, \theta) + \gamma \|\theta\|^2$, $\mathcal{B}$ represents the minibatch and $\gamma$ represents the weight decay. From a dynamical-system perspective, for an infinitesimal learning rate, the loss function coincides with the Bregman Lagrangian of this dynamics, and so one can leverage the Lagrangian formalism to understand the training of gradient flow [35]. This is particularly attractive from a theory perspective because modern theoretical physics are also founded on the Lagrangian formalism and this connection allows one to borrow physics intuitions to understand deep learning.

However, simply studying this loss function is insufficient to understand the learning dynamics of SGD at various learning rates $\eta$ due to the stochastic discrete-time nature of SGD. This motivates the definition of an entropic loss $\phi_\eta$ such that running $n$ steps of update on $\phi_\eta(\theta)$ with learning rate $\eta/n$

is the same as running one step of update on $\ell_\gamma$ for any $x$. Taking the limit $n \to \infty$, one can obtain a "renormalized" loss function for which running gradient flow is the same as running gradient descent for the original loss. With this loss, it becomes possible again to leverage Lagrangian formalism to understand SGD training with discrete-time and stochastic learning. Because $\phi_0$ coincides with running gradient flow on $\ell_\gamma$, one must have that

$$\phi_\eta := \ell_\gamma + \eta\phi_1 + \eta^2\phi_2 + O(\eta^3). \tag{2}$$

We can also consider the more general case where the learning rate is a fixed symmetric matrix $\Lambda$ with $\|\Lambda\| = \eta$. The following theorem derives the entropic loss for this case. Many common algorithms, such as Adam, natural gradient descent, and even a wide range of biologically plausible learning rules [36] can be seen as having matrix learning rate.

**Theorem 1.** *(Entropic Loss) For fixed $x$, starting from $\theta_0$ run one-step gradient descent with $\Lambda$ on $\ell_\gamma(x, \theta)$ to obtain $\theta_1$. Run $n$−step gradient descent with $\Lambda/n$ on $\phi_\Lambda(x, \theta) := \ell_\gamma(x, \theta) + \phi_{1\Lambda}(x, \theta) + \phi_{2\Lambda}(x, \theta)$ to obtain $\theta'_n$. Then, assuming $\|\nabla^3\ell_\gamma(x, \theta)\| \leq M$,*

1. *if $\phi_{1\Lambda}(x, \theta) = \frac{1}{4}\nabla\ell_\gamma(x, \theta)^T\Lambda\nabla\ell_\gamma(x, \theta)$, then, $\theta'_n = \theta_1 + O(\|\Lambda\|^3 + \|\Lambda\|^2/n)$;*
2. *moreover, if $\phi_{2\Lambda}(x, \theta) = \frac{1}{2}\nabla\ell_\gamma(x, \theta)^T\Lambda\nabla^2\ell(x, \theta)\Lambda\nabla\ell_\gamma(x, \theta)$, then $\theta'_n = \theta_1 + O(\|\Lambda\|^4 + \|\Lambda\|^2/n + \|\Lambda\|^3/n + \|\Lambda\|^3 M)$.*

This $\phi_\eta$ also needs to hold in expectation with respect to the sampling of data points, and so one can define the expected entropic loss $F_{\eta,\gamma}(\theta) = \mathbb{E}[\phi_\eta(x, \theta)]$, which is, up to the first order in $\Lambda$ and $\gamma$:

$$F_{\eta,\gamma}(\theta) = \underbrace{\mathbb{E}_x\ell(x, \theta)}_{\text{learning, symmetry}} + \underbrace{\gamma\|\theta\|^2}_{\text{regularization}} + \underbrace{\frac{1}{4}\mathbb{E}_\mathcal{B}\|\sqrt{\Lambda}\mathbb{E}_{x\in\mathcal{B}}\nabla\ell(x, \theta)\|^2}_{\text{effective entropy due to discretization error, noise, } := S(\theta)} + O(\|\Lambda\|^2). \tag{3}$$

Specializing to the first order and with a scalar learning rate, this equation reduces to the "modified loss" previously derived in different contexts [22, 23, 24]. The derivation has a thermodynamic flavor as it essentially computes the degree to which the dynamics is irreversible, and the entropy term is the part that cannot be microscopically reversed. In Theorem 1, the first-order term in $\Lambda$ encourages the model to have a small gradient fluctuation. The second-order term, on top of gradient regularization, encourages the model to move to flatter solutions; this term has been found to play a role in the edge of stability phenomenon [5]. However, since the first-order term is not yet well-understood, our work focuses on the first-order term in $\Lambda$. We also focus on a scalar learning rate $\Lambda = \eta I$ and will comment on the differences when the difference is essential.

Now, treating the original loss plus regularization as an energy, the dynamics of gradient flow on $F$ contains an energy force and and an entropic force: $\dot\theta = -\eta(\nabla L + \gamma\theta + \nabla S)$. The $\nabla S$ term, the gradient of the effective entropy, will be called the "entropic force." See Figure 1 for an illustration of the entropic force and an example of how entropy evolves during training.

We prove that the entropic force term breaks almost any continuous symmetry of $L$, a key result that we will leverage to study progressive sharpening and universal representation learning.

**Definition 1.** *A loss function $\ell(x, \theta)$ is said to be $K$-invariant if items 1-3 are satisfied:*

1. *locality: $K(\theta, \lambda) = \theta + \lambda Q(\theta) + O(\lambda^2)$ for a differentiable $Q$;*
2. *consistency: $K(K(\theta, \lambda), \lambda') = K(\theta, \lambda + \lambda')$;*
3. *invariance: $\ell(x, K(\theta, \lambda)) = \ell(x, \theta)$ for all $x$, $\theta$ and $\lambda \in \mathbb{R}$.*

*An entropic loss $F_{\eta,\gamma}$ is said to have the (**robust**) $K$-invariance if there exists a neighborhood around $\eta, \gamma$ such that $F_{\eta,\gamma}$ is $K$-invariant.*

**Theorem 2.** *(Symmetry Breaking Under the Entropic Loss) Let $\ell$ be $K$-invariant. If $F$ is also robustly $K$-invariant, then, (1) $\|K(\theta, \lambda)\| = \|\theta\|$ and (2) $\nabla^T\ell\nabla Q(\theta)\nabla\ell = 0$ for all $\theta$.*

This means that any symmetry or invariance that $F_{\eta,\gamma}$ has must be norm-preserving transformations. Essentially, this means that any invariance that is not rotation invariance must disappear. The following theorem shows that orthogonal discrete symmetries are preserved.

**Theorem 3.** *Let $OO^T = I$. If $\ell(x, O\theta) = \ell(x, \theta)$ for any $\theta$ and $x$, then $F_{\eta,\gamma}(O\theta) = F_{\eta,\gamma}(\theta)$ for any $\theta$.*

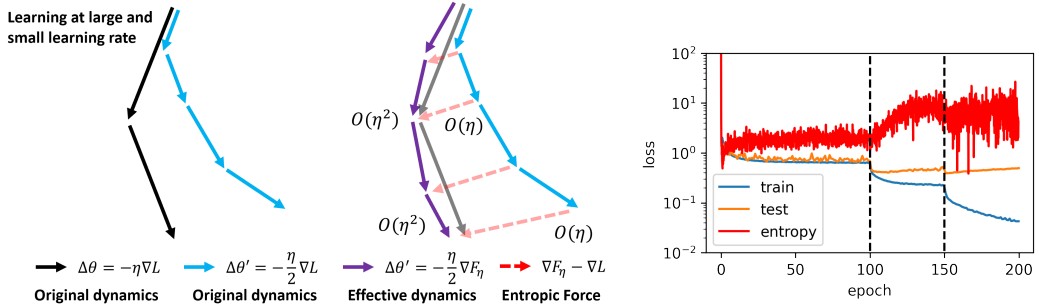

Figure 1: Entropic forces due to discretization error and stochasticity. **Left**: The learning dynamics of SGD at a large learning rate (LLR) and a small learning rate (SLR) is different. One can view the difference between LLR and SLR training as coming from an entropy term, which is an order $\eta$ force. After entropic correction, the difference between SLR and LLR is reduced to $O(\eta^2)$ and it becomes possible to analyze LLR SGD with gradient flow. **Right**: An example of entropic effect in neural network training. ResNet18 trained on CIFAR-10 with learning rate decay at 100 and 150 epochs. At the first learning rate drop (black dashed lines), the gradient (entropy) increases. This is unexpected and can only be explained by the entropic loss, where a large learning rate penalizes the entropy, and thus decreasing the learning rate leads to an increase in entropy. The second drop does not create too much effect because the learning rate is too small after the first drop.

Together with the previous result, this shows that when gradient noise or regularization is taken into account, the only relevant remaining symmetries are discrete symmetries. This implies that the results that are based on conservation laws for understanding SGD are questionable and can only hold in the toy setting of an infinitesimal learning rate. The reason is simple: $F_{\eta,\gamma}$ does not have *robust* invariances at $\eta = \gamma = 0$. Also, note that had we used a generic matrix learning rate (e.g., with Adam), the orthogonal invariances would also be broken. The meaning of these discrete symmetries can be understood through a framework similar to that proposed in Ref. [7] and is left to a future work.

## 4 Emergence of Gradient Balance and Equipartition Property

Lie group symmetries exist abundantly in nature and in modern neural networks[1] [27, 28, 29, 7, 32]. In thermodynamics, the existence of symmetries is a crucial fact that leads to the emergence of hierarchical phenomena and phase transitions between them. In a sense, symmetry can be argued to be the "first-order" approximation of the level of hierarchies in the system [37]. The following theorem states that the entropic loss $F$ breaks any nontrivial noncompact Lie group symmetries (also known as exponential symmetries [29]). For formality, we say that $\ell$ (and $L$) has a $A$-**exponential symmetry** if for any $\lambda \in \mathbb{R}$, any $x$ and $\theta$, and any matrix $A$, $\ell(x, \theta)$ obeys $\ell(x, e^{\lambda A}\theta) = \ell(x, \theta)$.

**Theorem 4.** *(Master Balance Theorem) If $\ell(x, \theta)$ has an A-exponential symmetry, then any local minimum $\theta^*$ of Eq.(3) satisfies*

$$-\eta \mathbb{E}_{\mathcal{B}}[\mathbb{E}_{x \in \mathcal{B}}(\nabla_\theta \ell(x, \theta^*))]^T \tilde{A}[\mathbb{E}_{x \in \mathcal{B}} \nabla_\theta \ell(x, \theta^*)] + 4\gamma(\theta^*)^T \tilde{A}\theta^* = 0, \tag{4}$$

*where $\tilde{A} := \frac{A+A^T}{2}$. In addition, either (1) $F_{\eta,\gamma}(e^{\lambda A}\theta^*) = F_{\eta,\gamma}(\theta^*)$ for all $\lambda$, or (2) there exists no $\lambda \neq 0$ such that $e^{\lambda A}\theta^*$ is a local minimum.*

If we had chosen a symmetric $A$, we would have $\tilde{A} = A$. For an anti-symmetric $A$, this theorem is trivial, consistent with Theorem 2. Therefore, this theorem is a statement about non-compact Lie group symmetries that extends to infinity. The fact that every point is connected to a point that satisfies Eq (4) means that SGD can reach this condition easily. The meaning of this theorem is a general gradient balance and alignment phenomenon. When $\gamma = 0$, this equation states that the gradient along the positive spectrum of $\tilde{A}$ must balance with the gradient along the negative spectrum. When $\gamma \neq 0$, there is, additionally, a tradeoff between gradient balance and weight balance. We apply this result to various neural networks in this section.

Many works have shown that when training with weight decay or when the model has a small initialization, the weights of the layers become balanced, especially in homogeneous networks [38,

---

[1]See Ref. [31] for a detailed review.

39, 40]. Theorem 4 shows that the SGD in discrete-time or with stochasticity leads to a completely different bias where the gradient noise between all layers must be balanced.

**ReLU Layers**    Consider a deep ReLU network trained on an arbitrary task:

$$f(x) = W_D R_{D-1}...R_1 W_1 x, \tag{5}$$

where $R(x)$ is a piece-wise constant the zero-one activation matrix functioning as the ReLU activation. The entropy term can be written as

$$S(\theta) = \sum_{i=1}^{D} \text{Tr} \mathbb{E}[g_i g_i^\top], \tag{6}$$

where $g_i = \mathbb{E}_{x \in \mathcal{B}} \nabla_{W_i} \ell(\theta, x)$ and $\mathbb{E}$ is a shorthand of $\mathbb{E}_\mathcal{B}$. Namely, we can group the gradient covariance according to layer index $i$. The following theorem states that all layers must have a balanced gradient. The proof shows that while the learning term $L$ is invariant to a class of symmetry transformations, the entropy term is not – and this creates a systematic tendency for the parameters to reduce the entropy.

**Theorem 5.**  *(Layer Balance) For all local minimum of Eq.* (3)*,*

$$\eta(\mathbb{E}\text{Tr}[g_i g_i^T - g_j g_j^T]) = 4\gamma(\text{Tr}[W_i W_i^T - W_j W_j^T]). \tag{7}$$

Specifically, for $\gamma = 0$ we have gradient balance $\mathbb{E}\text{Tr}[g_i g_i^\top] = \mathbb{E}\text{Tr}[g_j g_j^\top]$. For $\eta = 0$, we have the standard weight balance $\text{Tr}[W_i W_i^T] = \text{Tr}[W_j W_j^T]$. Otherwise, the solution interpolates between gradient balance and weight balance.

Similarly, within every two neighboring layers, one can group the parameters into neurons:

$$\text{Tr}[g_i g_i^\top] + \text{Tr}[g_{i+1} g_{i+1}^\top] = \sum_j \left( \text{Tr}[g_{i,j,:} g_{i,j,:}^\top] + \text{Tr}[g_{i+1,:,j} g_{i+1,:,j}^\top] \right), \tag{8}$$

where $g_{i,j,:}$ is the incoming weights to the $j$-the neuron of the $i$-th layer, and $g_{i+1,:,j}$ is the outgoing weights of the same neuron. The gradient for each neuron must also be balanced because there is a rescaling symmetry in every neuron.

**Theorem 6.**  *(Neuron Balance) For all local minimum of Eq.* (3) *and any* $i, j$*,*

$$\eta \mathbb{E}\text{Tr}[g_{i,j,:} g_{i,j,:}^\top - g_{i+1,:,j} g_{i+1,:,j}^\top] = 4\gamma \text{Tr}[w_{i,j,:} w_{i,j,:}^T - w_{i+1,:,j} w_{i+1,:,j}^T]. \tag{9}$$

From a physics perspective, we have proved an equipartition theorem (ET). The elements in the matrix $\mathbb{E}[g_i g_i^T]$ can be seen as the temperature (or, the average energy) felt by each parameter. The trace $\text{Tr}\mathbb{E}[g_i g_i^T]$ thus gives the temperature of the layer. That different layers emerge to have the gradient second momentum is an explicit form of the ET and means that the entropy $S$ must be evenly spread out across every layer. Because the standard ET in physics is a property of thermal equilibrium, our result may be seen as an extension of the physical law to the out-of-equilibrium dynamics of learning. See Figure 2 for the emergence of layer and neuron balances in a ReLU network. We train on the MNIST dataset, but the labels are generated by a teacher ReLU network and trained with an MSE loss. Also, see Appendix A for examples of training trajectories and for an example with a self-attention net.

**Polynomial Network**    Now, consider the case where $R(h)$ is a diagonal matrix such that $R_{ii}(h) = h^d$ corresponds to a polynomial activation. This type of network is also a variant of homogeneous networks [38]. For these networks, one can show that the converge to a state where every layer's gradient norm is $d$ times that of the previous layer, leading to a gradient exploding or vanishing problem (See Appendix B.8).

**Self Attention**    Consider the case when a model has a generic form: $\ell(x, W, U) = \ell(x, WU)$, where $W$ and $U$ are matrices. The loss can contain other trainable parameters, which we ignore. Define $G_W = \mathbb{E}_{x \in \mathcal{B}} \nabla_W \ell(x, W, U)$, $G_U = \mathbb{E}_{x \in \mathcal{B}} \nabla_U \ell(x, W, U)$, and one can prove the following relation:

**Theorem 7.**  *(Gradient Alignment) For all local minimum of Eq.*(3)*, we have*

$$\eta \mathbb{E}[G_W^T G_W - G_U G_U^T] = 4\gamma(W^T W - UU^T). \tag{10}$$

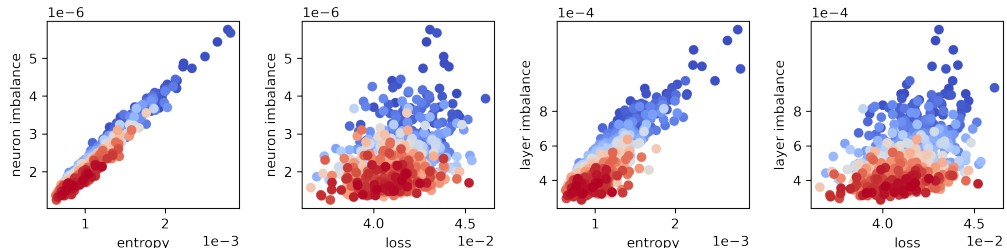

Figure 2: Layer and neuron gradient balance during training of a two-layer ReLU network. Here, every dot is a fixed time during training, where bluer dots are closer to the initialization, and redder dots are closer to convergence. **Left 1-2**: The entropy is strongly correlated with the neuron balance. As entropy decreases, the neuron balance improves. In contrast, the loss is not correlated to entropic effects at all. **Right**: Similarly, the layer balance is also correlated with entropy and not with loss function value.

This theorem is thus applicable to matrix factorization, deep linear networks, and, more importantly, self-attention layers. The self-attention logit is computed as $a_{ij} = X_i^T WUX_j$, where $W$ is the key matrix, $U$ is the query matrix. The loss function is a function of $a_{ij}$ viewed as a matrix: $\ell(\{a_{ij}\})$. Let $V = W_2 W_1$. Then, applying this theorem reveals an intriguing relation:

$$W_1 \mathbb{E}[G_V^T G_V] W_1^T = W_2^T \mathbb{E}[G_V G_V^T] W_2. \tag{11}$$

**Interpolating Weight Balance and Gradient Balance** For all the theorems above, we have also studied how weight decay affects the balance conditions. We see that the weight decay creates something analogous: instead of gradient balance, weight decay encourages weight balance, and this effect often cannot be achieved together with gradient balance. Thus, there is a tradeoff between gradient balance and weight balance. In reality, the network is somewhere in between, where the weight balance and gradient balance has to "balance" with each other. Also, this is a generalized form of an equipartition theorem. If we regard the sum of the regularization and $S$ as a "total" entropic potential $\Gamma$, then this means that every layer will contribute an equal amount to $\Gamma$.[2]

## 5 Implications

Next, we apply these results to study the emergence of universal representations in neural networks, and the progressive sharpening phenomena in deep learning optimization.

### 5.1 Universal Representation Learning

Recent works found that the representations of learned models are almost universally aligned to different models trained on similar datasets [41, 42], and even to the biological brains [43]. This interesting phenomenon has a rather philosophical undertone and has been termed "Platonic Representation Hypothesis" [8]. Here, we say that the two neural networks have learned a universal representation if for all $x_1$, $x_2$,

$$h_A(x_1)^T h_A(x_2) = h_B(x_1)^T h_B(x_2), \tag{12}$$

where $h_A$ is the activation of network $A$ in one of the hidden layers, and $h_B$ for network B. This is an idealization of what people have observed – and the difference between the two sides is the "degree of alignment." We leverage the entropic force formalism to identify an exact solution to the embedded deep linear network (EDLN) model:

$$\ell(\theta, x) = \|M_1 W_D \cdots W_1 M_2 x - y(x)\|^2. \tag{13}$$

on datasets $\mathcal{D}_{M_3} = \{(M_3 x_i, y_i)\}_i$, where $M_1, M_2, M_3$ are fixed but arbitrary invertible matrices. They have the following meaning:

- $M_1$ can be seen as a model of the layers coming after the deep linear network;

---

[2]Now, it is helpful to clarify the role of the $L_2$ regularization term. Conceptually, it can either be regarded as a part of the loss function $\ell$, which we have done so far, or as a part of the entropic term. Treating it as a part of entropy is sometimes conceptually preferred because, like the entropy, it also functions as a regularization of the parameter space, limiting the accessible states.

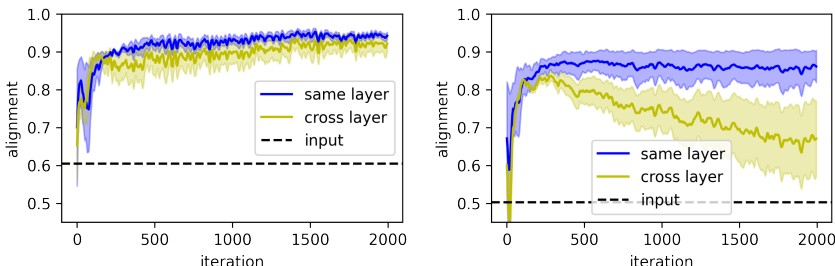

Figure 3: The representations of two 6-layer networks independently trained on randomly transformed MNIST become perfectly aligned for **every** pair of layers. The figure shows the average alignment between the same or different layers of two networks. This alignment does not weaken even if the input is arbitrarily transformed (Theorem 8). The black dashed line shows the average alignment to the input data, which is significantly weaker. **Left**: linear network. **Right**: tanh network.

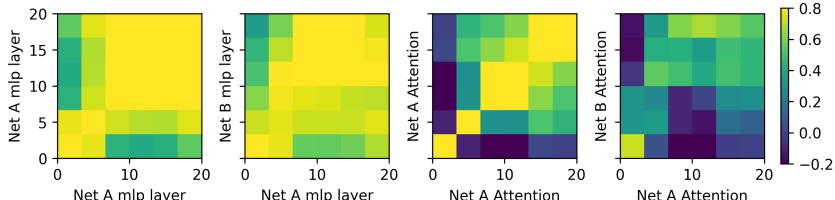

Figure 4: Alignment of representations of two ViT models pretrained on ImageNet. Net A: ViT-B (#param: 86M). Net-B: ViT-H (300M). We see that both mlp layers and the self-attention layers have mutually aligned representations both with itself and the other with each other. In particular, the alignment with itself is slightly better than with the different model, and the alignment of later layers is better than that of the first layers. A similar result with larger models is shown in Appendix A.5.

- $M_2$ models the layers coming after the embedded network;
- $M_3$ models different views of the data, which is common in multimodal learning – therefore, the two models are trained on two different (but related) datasets.

In Theorem 8, we will train two different models, each with their own and potentially different $M_1$, $M_2$, $M_3$. The arbitrariness of these three matrices implies *universality*.

The data is generated by $y_i = V x_i + \epsilon_i$ for i.i.d. noise $\epsilon_i$. Assuming that $\mathbb{E} x_i = \mathbb{E} \epsilon_i = 0$ and $\Sigma_\epsilon := \mathbb{E} \epsilon_i \epsilon_i^T$, $\Sigma_x := \mathbb{E} x_i x_i^T$, the following theorem characterizes the global minimum of the entropic loss for this network in the case $\gamma = 0$ and $\eta \to 0_+$.

**Theorem 8** (Perfect Platonic Representation Hypothesis). *Consider two deep linear networks A and B with weights of arbitrary dimensions larger than $rank(\sqrt{\Sigma_\epsilon} V \sqrt{\Sigma_x})$. Let model A train on $\mathcal{D}_{M_3}$ and model B on $\mathcal{D}_{M_3'}$. Moreover, the outputs of models are multiplied by $M_1, M_1'$, and the inputs are multiplied by $M_2, M_2'$, respectively. Then, at the global minimum of Eq. (3), every hidden layer of A is **perfectly** aligned with every hidden layer of B for any x, in the sense that*

$$h_A^{L_A}(x) = c_0 R h_B^{L_B}(x) \tag{14}$$

*for $1 \le L_A < D_A$ and $1 \le L_B < D_B$ and any x, where $c_0$ is a scalar and $R = U_1 U_2^T$ satisfying $U_1^T U_1 = U_2^T U_2 = I$. $h_A^{L_A}(x) := \Pi_{i=1}^{L_A} W_i^A M_2 M_3 x$, $h_A^{L_A}(x) := \Pi_{i=1}^{L_A} W_i^A M_2' M_3' x$ denote the output of the $L_A, L_B$−the layer of network A and B, respectively.*

Here, a perfect alignment means that $h_A^{L_A}(x)$ differs from $h_B^{L_B}(x)$ only by a scaling and a rotation. Because of symmetry, SGD converges to a state where all possible pairs of the intermediate layers of two different networks are mutually aligned, independent of the initialization. This is an extraordinary fact because there exist infinitely many solutions that are not perfectly aligned, also due to symmetry. For example, take $h_B$ to be any hidden layer, and transform its incoming weight by $A$ and its outgoing weight by $A^{-1}$. This remains a global minimum for $L$, but it is no longer the case that there exists an orthogonal transformation $R$ such that $h_A = h_B$. Therefore, for almost all global minima of $L$, there is no universal alignment between layers – yet, SGD prefers a universal solution due to the entropy term. The following theorem shows that weight decay will lead to a nonuniversal representation.

From a physics and thermodynamics perspective, it is quite reasonable that the irreversibility of the dynamics leads to the emergence of universal structures. A state is not universal if it contains information about its initial condition. Therefore, the irreversibility of the learning dynamics helps erase information about the parameter initialization, thereby enabling the learning of a universal solution. The following hypothesis can summarize this perspective:

*Irreversibility enables universal representation learning.*

Now, it is worthwhile to remark on the connection and difference between this result and the original PRH [8]. First, our theory provides strong support for the PRH. The original PRH only hypothesizes a positive similarity between models, and it is unclear whether the alignment score can reach 1 (which implies a perfect alignment) or will only be a small positive value. Our result shows that, in principle, it is possible to reach the perfect alignment limit. Secondly, this result also offers an alternative perspective on the representation alignment phenomenon to that of the original PRH paper. In the old perspective, one regards having no alignment as the default expectation, and positive alignment as something to be explained and understood. In our new perspective, the perfect alignment is the default, and breaking away from it is something to be explained and understood – which is exactly what we will demonstrate in the next theorem. We also refer to Ref. [44] for a more instructive derivation of the proof and for a detailed discussion of different ways to break the perfect PRH.

**Theorem 9.** *Consider the deep linear network* (13) *with widths larger than* $d := rank(V)$. *Let* $\eta = 0$ *and* $\gamma \to 0_+$. *At the global minimum of* (3)*, we have*

$$W_i = U_i P_i \Sigma U_{i-1}^T \tag{15}$$

*for* $i = 1, \cdots, D$. $U_D$ *and* $U_0$ *are given by the SVD of* $M_1^{-1} V M_3^{-1} M_2^{-1} := U_D S U_0$*, where* $S \in \mathbb{R}^{d \times d}$ *contains the singular values. For* $i = 2, \cdots, D - 1$*,* $U_i$ *are arbitrary matrices satisfying* $U_i^T U_i = I_{d \times d}$. *Moreover,* $\Sigma = S^{1/D}$. $\{P_i\}_{i=1}^D \subset \mathbb{R}^{d \times d}$ *are diagonal matrices containing* $\pm 1$ *and satisfy* $\Pi_{i=1}^D P_i = I_d$.

The universal representation property (Theorem 8) does not hold anymore. Therefore, gradient balances lead to universal representations, whereas weight balances do not. See Figure 3 for an experiment with deep linear and nonlinear networks. A surprising aspect is that every layer can be aligned with every other layer. Because any deep nonlinear network is approximated by a linear network for a small weight norm [45], one could say that any nonlinear network is, to first order in $\|\theta\|$, a universal representation learner. In Figure 4, we compare the alignment between different self-attention layers of two differently sized vision transformers pretrained on ImageNet. Note that different layers of the same network and different networks also have significantly positive alignments, consistent with the solution for the embedded deep linear net. That using weight decay does not lead to universal representations is supported by the result in Figure 10.

Theorem 8 is a direct (perhaps the first) proof of the Platonic representation hypothesis, implying that for any $x_1, x_2$, Eq. (12) holds. Importantly, the mechanism does not belong to any previously conjectured mechanisms (capacity, simplicity, multitasking [8]). This example has nothing to do with multitasking. The result holds for any deep linear network, all having the same capacity and the same level of simplicity, because all solutions parametrize the same input-output map. Here, the cause of the universal representation is symmetry alone: in the degenerate manifold of solutions, the training algorithm prefers a particular and universal one. This example showcases how symmetry is indeed an overlooked fundamental mechanism in deep learning.

## 5.2 The Sharpness Paradox

The tendency towards learning universal representations can imply a curse for training. For example, the sharpness of the loss landscape really depends on the distribution of the data, whereas the solution the model finds is independent of these distributions – this could imply that these solutions can be quite bad in terms of, say, optimization properties. Meanwhile, a paradox of the sharpness-seeking behavior of SGD has become explicit. On the one hand, the edge of stability (EOS) states that learning *typically* leads to sharper solutions, whereas a vast majority of works have shown that SGD training leads to flatter solutions [46, 47]. These two cannot happen simultaneously – and the solution must be that the sharpness-seeking behavior of SGD is situation-dependent. This section formalizes this intuition and shows that universal representation learning can be intrinsically related to the edge of stability phenomenon, which is also ubiquitous in deep learning.

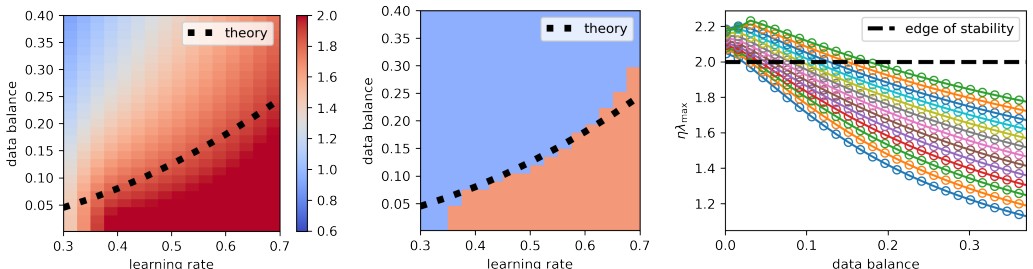

Figure 5: The entropic theory predicts the boundary for the edge of stability (EOS) phenomenon [4]. The theory shows that the imbalance of features and the uncertainty of labels make the model converge to sharper solutions. We run a two-layer linear network trained on a regression task. The **Left** panel plots the quantity $\eta\lambda_{\max}$ at convergence. For stability, $\eta\lambda_{\max}$ must stay (approximately) below 2, and the black dotted line plots the theoretical boundary for $\eta\lambda_{\max} = 2$. **Middle**: The same figure that emphasizes the phase boundary. The blue-red boundary empirically defined by the condition $\eta\lambda_{\max} = 2 - \epsilon$ with $\epsilon = 0.1$ – due to random sampling, the actual edge of stability is slightly smaller than 2 [20]. **Right**: We control the learning rate and balance of the label noise for this experiment. As predicted, as the data noise becomes more balanced, the sharpness metric $\eta\lambda_{\max}$ gets smaller, indicating better dynamical stability during training.

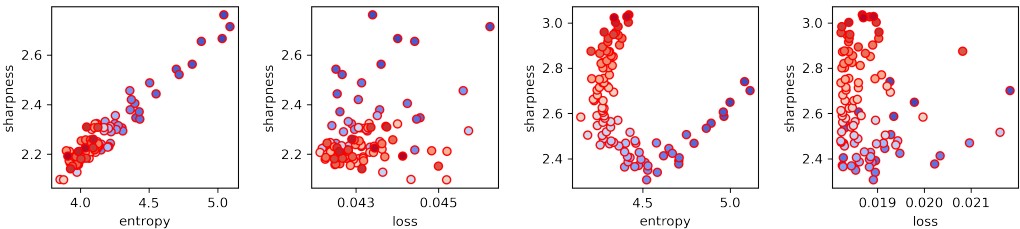

Figure 6: Example of a trajectory of training during the progressive flattening (**Left 1, 2**) and sharpening (**Right 1, 2**) of the experiment in Figure 5. Here, blue dots correspond to earlier in the training, and red dots correspond to later in the training. During progressive flattening, the decrease in sharpness correlates directly with the entropy term (Left 1), whereas the training loss is independent of the sharpness (Left 2). For progressive sharpening (Right), the picture is more complicated. The training trajectory follows three phases. Phase I: flattening correlates with a decrease in entropy; phase II: sharpening correlates with a decreasing entropy; phase III: sharpening correlates with an increasing entropy. The phase III cannot be explained by the leading-order entropic loss because, as the landscape becomes sharper, higher-order effects in $\eta$ start to dominate training. At the same time, the loss is never correlated with these effects (Right 4).

**Definition 2.** *The total sharpness is defined as $T(\theta) = \mathrm{Tr}\mathbb{E}\nabla^2\ell(x,\theta)$.*

This definition is chosen for analytical tractability and has been used in prior works [48, 29]. $T$ upper bounds the largest eigenvalue, and $T/d$ lower bounds it, so it is a good metric of stability and sufficient for the theorem we will prove. The following lemma shows that if there is an exponential symmetry in $\ell$, every local minimum of $\ell$ connects without barrier to a local minimum with arbitrarily high sharpness. It can be seen as a generalization of the result of Ref. [49] to general symmetries.

**Lemma 1.** *(Sharpness Lemma) Assume that $A$ is a symmetric matrix and $\ell(x, e^{\lambda A}\theta) = \ell(x,\theta)$. Moreover, assume that $A\mathbb{E}\nabla^2\ell(x,\theta) \neq 0$. Then, $\limsup_{|\lambda|\to+\infty} |T(e^{\lambda A}\theta)| = +\infty$.*

One can analytically solve for the sharpness of two-layer linear networks, and identify a precise cause of the progressive sharpening effect.

**Theorem 10.** *For a two-layer linear network $\ell(x,\theta) = \|y(x) - W_1W_2x\|^2$ with $y(x) = Vx + \epsilon \in \mathbb{R}^{d_y}$ and $\mathbb{E}x = \mathbb{E}\epsilon = 0$, $\Sigma_x = \mathbb{E}xx^T$, $\Sigma_\epsilon = \mathbb{E}\epsilon\epsilon^T$. Denote $\tilde{U}S'\tilde{V}$ to be the SVD of $V' := \sqrt{\Sigma_\epsilon}V\sqrt{\Sigma_x}$ and assume that the width of the network is larger than $\mathrm{rank}(V')$. Then we have*

$$T(\theta) = d_y\sqrt{\frac{\mathrm{Tr}[\Sigma_x]}{\mathrm{Tr}[\Sigma_\epsilon]}}\mathrm{Tr}[S'] + \sqrt{\mathrm{Tr}[\Sigma_x]\mathrm{Tr}[\Sigma_\epsilon]}\mathrm{Tr}[\Sigma_\epsilon^{-1}\tilde{U}S'\tilde{U}^T] \qquad (16)$$

*at the global optimum of* (3). *Meanwhile, the minimal sharpness of the global minimum of $\ell$ is*

$$\min T(\theta) = 2\sqrt{d_y\mathrm{Tr}\Sigma_x\mathrm{Tr}\hat{S}}, \qquad (17)$$

*where $\hat{S}$ is the singular values of $V\sqrt{\Sigma_x}$.*

This result implies that SGD has no inherent preference for flatter minima. See Figure 5-6, where we train a two-layer linear network on a linear regression task with a 2d label $y \in \mathbb{R}^2$. The labels $y = V^*x + \epsilon$, for a ground truth matrix $V^*$ and iid zero-mean noise $\epsilon$ such that $\Sigma_\epsilon = \text{diag}(1, \phi_x)$, where $\phi_x \in (0, 1)$ is called the "data balance." In Appendix A.6, we also train a deep nonlinear network, and we see the same trend where improving balance in the label noise leads to flatter solutions.

As an example, we can choose $\Sigma_x = I$, $V = I$ ($d_x = d_y = d$), which gives $V' = \sqrt{\Sigma_\epsilon} = \tilde{U}S'\tilde{U}^T$, and thus

$$T(\theta) = d^{3/2}\text{Tr}[\Sigma_\epsilon]^{-1/2}\text{Tr}[\Sigma_\epsilon^{1/2}] + d^{1/2}\text{Tr}[\Sigma_\epsilon]^{1/2}\text{Tr}[\Sigma_\epsilon^{-1/2}], \tag{18}$$

which can be arbitrarily large. Recall that the minimum of $T(\theta)$, on the other hand, does not depend on $\Sigma_\epsilon$, the label noise covariance. Thus, the imbalance of the noise spectrum can lead to arbitrarily high sharpness. This could especially be a problem for language model training because there is a large variation in the randomness of tokens. Some words, like "the," could have a very low conditional entropy, while nouns or verbs can have high entropy, especially when there exist synonyms. Another example is to choose $\Sigma_\epsilon = I$, which gives $V' = V\sqrt{\Sigma_x} = \tilde{U}S'\tilde{U}^T$, and thus

$$T(\theta) = 2\sqrt{d}\text{Tr}[\Sigma_x]^{1/2}\text{Tr}[V\Sigma_x^{1/2}], \tag{19}$$

which is exactly the same as the minimal sharpness. This suggests that without the imbalance of the label noise alone, SGD indeed converges to the flattest solution.

Thus, an imbalance in the input feature can lead to different sharpness-seeking behaviors. At the same time, if the loss function has scale invariance ($\ell(x, \theta) = \ell(x, \lambda\theta)$ for any $\lambda \in \mathbb{R}$), the learning dynamics leads to a flattening of the curvature during training (See Section B.1). Thus, entropic forces and symmetry are strong factors deciding the sharpness-seeking behavior of SGD.[3]

# 6    Conclusion

In this work, we have proposed an entropic-force perspective on neural network learning. We derive an entropic loss, which breaks continuous symmetries and preserves discrete ones, leading to universal behaviors such as gradient balance and alignment. The entropic loss suggests a potential unifying perspective for understanding training: learning algorithms prefer solutions with the minimal gradient fluctuation. This perspective provides a unifying framework that explains several emergent phenomena in deep learning, including sharpness-seeking behavior and universal feature structure, as consequences of underlying symmetries and entropic forces. The framework offers predictive and explanatory power across architectures and scales, and points toward a more principled, physics-inspired understanding of learning dynamics and emergent phenomena. Future work may extend this foundation to encompass higher-order corrections, richer architecture structures, and nonequilibrium dynamics of modern training procedures. A major limitation of our work is that we focused only on problems with explicit symmetries; an important future direction is to extend the results to cases with only approximate symmetries. Our experiments show that symmetry-free systems qualitatively agree with symmetry-preserving systems, suggesting that there are underlying, hidden concepts yet to be discovered.

An implication of Theorem 2 is that discrete symmetries such as $Z_2$ still remain in the loss function. This enables the possibilities of spontaneous symmetry breaking and phase transitions in neural networks. Prior works have studied phase transitions [7] without the entropy term and are inherently zero-temperature phase transitions. It may be possible to develop a theory of phase transitions based on entropic loss, similar to the classical Landau theory. Also, a striking aspect of our construction is its similarity to the actual formalism of thermodynamics in physics; our result motivates the development of a robust thermodynamics theory of deep learning.

---

[3]Viewed together with the result in Section 5, one reaches an interesting and surprising conjecture: progressive sharpening and universal representation alignment, with entirely different phenomenology, may have the same underlying cause, and could be two sides of the same coin.

## Acknowledgment

The authors thank Prof. Phillip Isola and Prof. Tomaso Poggio for discussion. ILC acknowledges support in part from the Institute for Artificial Intelligence and Fundamental Interactions (IAIFI) through NSF Grant No. PHY-2019786. This work was also supported by the Center for Brains, Minds and Machines (CBMM), funded by NSF STC award CCF - 1231216.

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

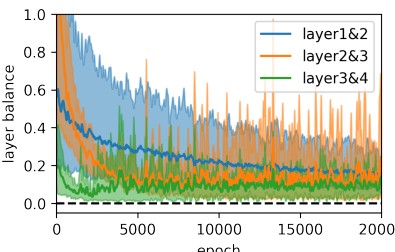 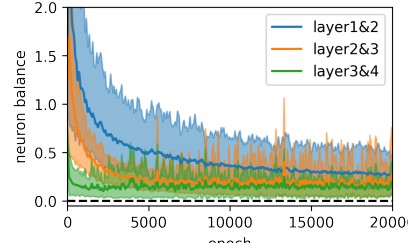

Figure 7: 4-layer linear networks and no weight decay trained on a teacher-student setting. **Left**: Layer imbalance for each layer, which verifies Theorem 5. **Middle**: Neuron balance for each layer, which verifies Theorem 6. The curves are smoothed and averaged over 5 runs for better visualization. **Right**: Loss and entropy.

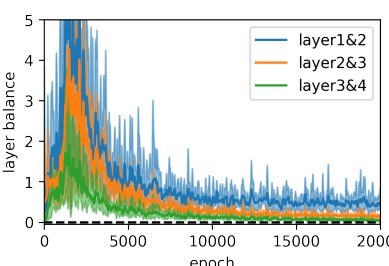 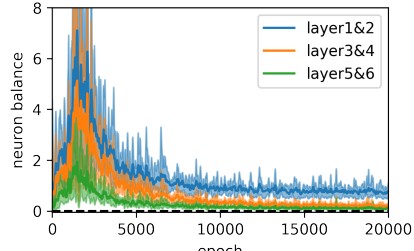

Figure 8: The same setting as Figure 7, but for ReLU activation.

## A Experiment

### A.1 ResNet

For Figure 1, we train ResNet18 on CIFAR 10 using SGD with momentum $0.9$, batchsize $128$ and weight decay $5 \times 10^{-4}$. The learning rate is $0.1$ at the beginning, $0.01$ after the $100$−th epoch and $0.001$ after the $150$−th epoch. The entropy is calculated by summing the gradient norm of all parameters. We obtain training accuracy $98\%$ and test accuracy $88\%$ at the end.

### A.2 Gradient Balance

For Figure 2, we train on the MNIST dataset but the labels are generated by a teacher ReLU network and trained with an MSE loss. Namely, the loss is

$$\|f(\theta) - y(x) - \epsilon_x\|^2, \tag{20}$$

where $y(x)$ is a parameterized by a random ReLU teacher network and $\epsilon_x$ is an i.i.d. Gaussian noise with $0.2$ standard deviation. The training proceeds with SGD for $10^4$ steps with a learning rate of $0.01$ and batchsize of $200$. Layer balance is calculated by $|\mathbb{E}\mathrm{Tr}[g_i g_i^T - g_{i+1}^T g_{i+1}]|$ and neuron balance is calculated by $\sum_j |\mathbb{E}\mathrm{Tr}[g_{i,j,:} g_{i,j,:}^\top - g_{i+1,:,j} g_{i+1,:,j}^\top]|$ for the $i$−th layer.

Additional experiments on layer balance and neuron balance are presented in Figures 7, 8 and 9 for 4-layer linear networks, ReLU networks and simple self-attention networks, still for the teacher-student setting. The hidden dimensions are $256, 128, 64$ for linear and ReLU networks, and $256$ for self-attention. We present the evolution of layer and neuron balance along training. Figures 7 and 8 suggest that layer balance and neuron balance approach zero during training, which verifies Theorems 5 and 6.

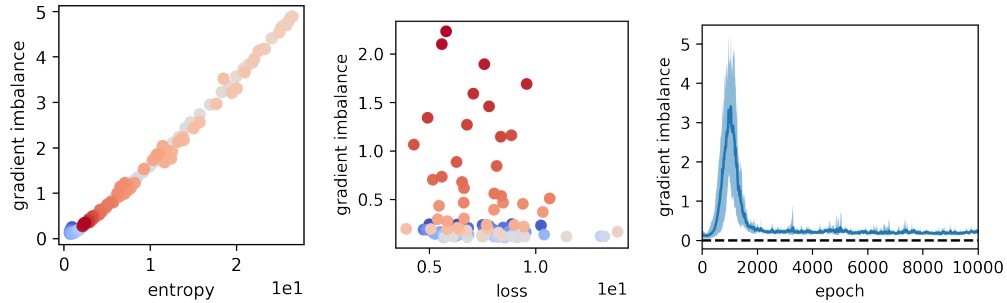

Figure 9: A self attention networks $y = (x^T U V x) w^T x$ and no weight decay trained on a teacher-student setting. The gradient imbalance is evaluated by $\|G_U^T G_U - G_V G_V^T\|_F$. **Left**: Gradient imbalance is strongly correlated with the entropy. **Middle**: Gradient imbalance is weakly correlated with the entropy. **Right**: The evolution of entropy along training, which is averaged over 5 runs.

## A.3 Gradient Balance in Self-Attention Nets

Figure 9 suggests that $G_U^T G_U - G_V G_V^T$ approaches zero during training, and it is correlated with the entropy rather than the loss, which verifies Theorem 7.

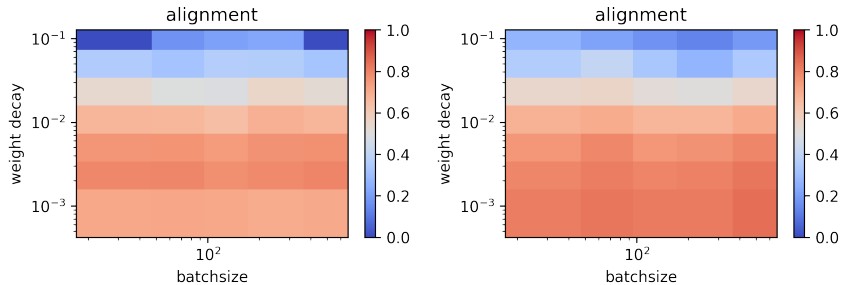

Figure 10: The alignment of two 2−layer ReLU networks independently trained on a teacher-student setting. We measure the alignment for different batchsizes and weight decay. **Left**: SGD. **Right**: Adam.

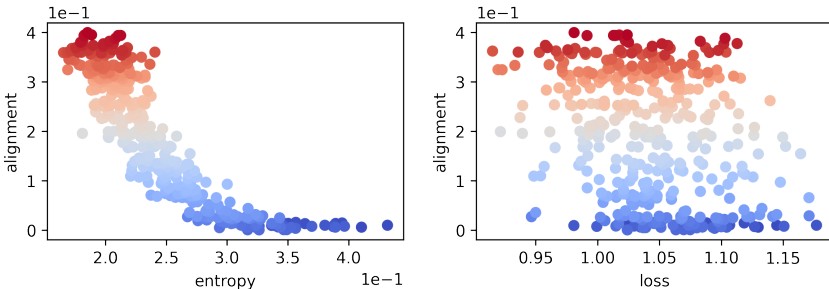

Figure 11: Universal alignment is strongly correlated with the entropy but not the loss. The setting is the same as Figure 10, where we use SGD, weight decay 0 and batchsize 100.

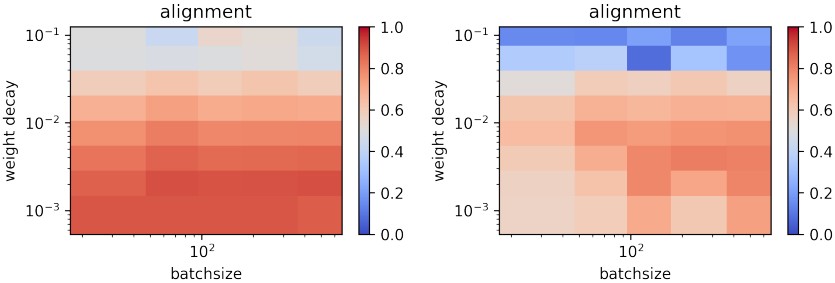

Figure 12: The same setting as Figure 10, but for two-layer linear networks. **Left**: SGD. **Right**: Adam.

## A.4 Unversal Representation Learning in MLP

For Figure 3, we train two independent 6-layer networks on MNIST. The networks have linear or tanh activation and 128 neurons in each hidden layer, and for the second network, the input MNIST data is transformed by a random Gaussian matrix. We train the networks with Adam optimizer, learning rate $10^{-4}$ for 5 epochs. During training, we measure the representation alignment between every pair of layers, defined as the cosine similarity between the two sides of (12), averaged over the test set. We then plot the average alignment between the same or different layers of two networks. The input alignment denotes the average alignment between every layer representation and the input data.

In Figures 10 and 12, we test the influence of batchsize and weight decay on universal representation in a teacher-student setting. In Figure 10, both the teachers and students are two-layer ReLU networks. In Figure 12, the student is replaced by a linear network. Their hidden dimensions are 100. Similar to Figure 3, we measure the representation alignment between the middle layers of two independently trained networks, and the input of the second network is rotated by a Gaussian matrix. We use random Gaussian data, and the labels are generated by the teacher network. We train the student networks with SGD, learning rate $5 \times 10^{-2}$ or with Adam, learning rate $10^{-4}$. For both SGD and Adam optimizers, Figures 10 and 12 suggest that universal alignment does not rely on the batchsize as predicted, but disappears for large weight decay, which verifies Theorem 9. In Figure

[11](#), we show that the increase of alignment is more correlated with the decrease of the entropy rather than the loss.

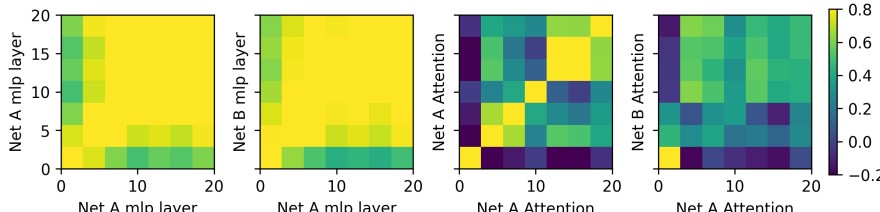

Figure 13: Alignment of representations of two larger ViT models pretrained on ImageNet. Net A: ViT-L (#param: 304M). Net-B: ViT-H (633M). This is similar to Figure 4

## A.5 Universal Representation in ViT

See Figure 13 for the alignment in Vision Transformer. The pretrained weights are taken from https://docs.pytorch.org/vision/main/models.html. We measure the CKA alignment between the two models or with itself with a minibatch size of 300 images from the ImageNet dataset.

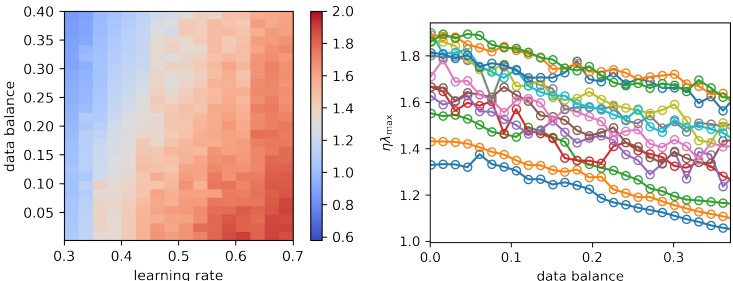

Figure 14: Sharpness at convergence for a two-hidden-layer ReLU network. The setting is identical to that of Figure 5. Again, we see that higher imbalance in the label leads to a sharper solution.

## A.6 Edge of Stability

See Figure 5 and 6, where we train a two-layer linear network on a linear regression task with a 2d label $y \in \mathbb{R}^2$. The labels $y = V^* x + \epsilon$, for a ground truth matrix $V^*$ and iid zero-mean noise $\epsilon$ such that $\Sigma_\epsilon = \mathrm{diag}(1, \phi_x)$, where $\phi_x \in (0, 1)$ is called the "data balance."

We train with different learning rates and data balance. The training proceeds with SGD with a batchsize of 32 for $4 \times 10^4$ iterations. Input $x \in \mathbb{R}^2$ is drawn from a standard Gaussian distribution, and the model has dimensions $2 \to 10 \to 2$.

See Figure 14 for an experiment with ReLU. Here, the architecture is has dimensions $2 \to 10 \to 10 \to 2$.

## B Theory

### B.1 Scale Invariance Leads to Flattening

The symmetry generator is $A = I$. We have

$$\frac{d}{d\lambda} F_{\eta,0}(e^\lambda \theta) = -\eta \mathbb{E}_x (\nabla_\theta \ell(x,\theta)^T \nabla_\theta \ell(x,\theta)) = -S < 0 \tag{21}$$

by Theorem 4. As $\frac{d}{d\lambda} F_{\eta,\gamma}(e^\lambda \theta)|_{\lambda=0} = \theta^T \nabla F_{\eta,0}$. Therefore, when we do gradient descent along $-\nabla F_{\eta,0}$, $\lambda$ monotonously increases. Meanwhile,

$$T(e^\lambda \theta) = \text{Tr}[e^{-2\lambda} \mathbb{E} \nabla^2 \ell(x,\theta)] \tag{22}$$

decreases with $\lambda$, and thus the sharpness decreases along training.

### B.2 Proof of Theorem 1

*Proof.* For notational simplicity we drop the subscript $\gamma$ in the proof. When running gradient descent on $\ell(x,\theta)$, we have

$$\theta_1 = \theta_t - \Lambda \nabla \ell(x,\theta_0). \tag{23}$$

When running gradient descent on $\phi_\Lambda$, we have

$$\theta_1' = \theta_0 - \frac{\Lambda}{n}(\nabla \ell(x,\theta_0) + \nabla \phi_{1\Lambda}(x,\theta_0) + \nabla \phi_{2\Lambda}(x,\theta_0)) + O(\|\Lambda\|^3) \tag{24}$$

and

$$\begin{aligned}
\theta_2' &= \theta_1' - \frac{\Lambda}{n}(\nabla \ell(x,\theta_1') + \nabla \phi_1(x,\theta_1') + \nabla \phi_2(x,\theta_1')) + O(\|\Lambda\|^4) \\
&= \theta_0 - \frac{2\Lambda}{n} \nabla \ell(x,\theta_0) + \frac{\Lambda}{n^2} \nabla^2 \ell(x,\theta_0) \Lambda \nabla \ell(x,\theta_0) - \frac{\Lambda}{n} \nabla \phi_1(x,\theta_0) \\
&\quad + \frac{\Lambda}{n^2} \nabla^2 \phi_1(x,\theta_0) \Lambda \nabla \ell(x,\theta_0) - \frac{\Lambda}{n} \nabla \phi_2(x,\theta_0) + O(\|\Lambda\|^4).
\end{aligned} \tag{25}$$

Similarly we can obtain

$$\begin{aligned}
\theta_n' &= \theta_0 - \Lambda \nabla \ell(x,\theta_0) + \frac{\Lambda}{2} \nabla^2 \ell(x,\theta_0) \Lambda \nabla \ell(x,\theta_0) - \Lambda \nabla \phi_1(x,\theta_0) \\
&\quad + \frac{\Lambda}{2} \nabla^2 \phi_1(x,\theta_0) \Lambda \nabla \ell(x,\theta_0) - \Lambda \nabla \phi_2(x,\theta_0) + O(\|\Lambda\|^4 + \|\Lambda\|^2/n + \|\Lambda\|^3/n).
\end{aligned} \tag{26}$$

Therefore, we have $\theta_n' = \theta_1 + O(\|\Lambda\|^3 + \|\Lambda\|^2/n)$ if we choose $\phi_1(x,\theta) = \frac{1}{4} \nabla \ell(x,\theta)^T \Lambda \nabla \ell(x,\theta)$.

For small $\nabla^3 \ell(x,\theta)$, we have

$$\begin{aligned}
\nabla^2 \phi_1(x,\theta_0) \Lambda \nabla \ell(x,\theta_0) &= \nabla^2 \ell(x,\theta_0) \Lambda \nabla^2 \ell(x,\theta_0) \Lambda \nabla \ell(x,\theta_0) + O(\|\Lambda\|^2 \|\nabla^3 \ell(x,\theta)\|) \\
&= \frac{1}{2} \nabla[\nabla \ell(x,\theta_0)^T \Lambda \nabla^2 \ell(x,\theta_0) \Lambda \nabla \ell(x,\theta_0)] + O(\|\Lambda\|^2 \|\nabla^3 \ell(x,\theta)\|).
\end{aligned} \tag{27}$$

Therefore, we can choose

$$\phi_2(x,\theta) := \frac{1}{2} \nabla \ell(x,\theta)^T \Lambda \nabla^2 \ell(x,\theta) \Lambda \nabla \ell(x,\theta) \tag{28}$$

to obtain $\theta_n' = \theta_1 + O(\|\Lambda\|^4 + \|\Lambda\|^2/n + \|\Lambda\|^3/n + \|\Lambda\|^3 \|\nabla^3 \ell(x,\theta)\|)$. $\qquad\square$

### B.3 Proof of Theorem 4

*Proof.* By the definition of the exponential symmetry,

$$\ell(x, e^{\lambda A} \theta) = \ell(x,\theta), \tag{29}$$

Taking derivative w.r.t. $\lambda$ on (29), we have that $\nabla_\theta \ell(x,\theta)^T A\theta = 0$. Then taking derivative w.r.t. $\theta$, we have that $A^T \nabla_\theta \ell(x,\theta) + \nabla_\theta^2 \ell(x,\theta) A\theta = 0$.

Let $I(\lambda) \coloneqq \frac{d}{d\lambda} F_{\eta,\gamma}(e^{\lambda A}\theta^*)$ and $\theta_\lambda \coloneqq e^{\lambda A}\theta^*$. Then we have

$$
\begin{aligned}
I(\lambda) &= \frac{\eta}{2}(\theta_\lambda)^T A^T \mathbb{E}_{\mathcal{B}}[\mathbb{E}_{x\in\mathcal{B}}\nabla^2\ell(x,\theta_\lambda)][\mathbb{E}_{x\in\mathcal{B}}\nabla\ell(x,\theta_\lambda)] + 2\gamma(\theta_\lambda)^T A\theta_\lambda \\
&= -\frac{\eta}{2}\mathbb{E}_{\mathcal{B}}(\mathbb{E}_{x\in\mathcal{B}}\nabla_\theta\ell(x,\theta_\lambda))^T A\mathbb{E}_{x\in\mathcal{B}}\nabla_\theta\ell(x,\theta_\lambda) + 2\gamma(\theta_\lambda)^T A\theta_\lambda \\
&= -\frac{\eta}{2}\mathrm{Tr}[\Sigma(\theta_\lambda)A] + 2\gamma(\theta_\lambda)^T A\theta_\lambda \\
&= -\frac{\eta}{2}\mathrm{Tr}[\Sigma(\theta_\lambda)\tilde{A}] + 2\gamma(\theta_\lambda)^T \tilde{A}\theta_\lambda,
\end{aligned}
\tag{30}
$$

where $\Sigma(\theta_\lambda) \coloneqq \mathbb{E}_{\mathcal{B}}[\mathbb{E}_{x\in\mathcal{B}}\nabla_\theta\ell(x,\theta_\lambda)][\mathbb{E}_{x\in\mathcal{B}}\nabla_\theta\ell(x,\theta_\lambda)]^T$ is positive semi-definite. We also use $\mathrm{Tr}[\Sigma(\theta_\lambda)\frac{A-A^T}{2}] = 0$ and $(\theta_\lambda)^T \frac{A-A^T}{2}\theta_\lambda = 0$. By [29, Lemma B.1], we have $\mathrm{Tr}[\Sigma(\theta_\lambda)\tilde{A}] = \mathrm{Tr}[e^{-2\lambda}\Sigma(\theta^*)\tilde{A}]$, and thus

$$
\begin{aligned}
I(\lambda) &= -\frac{\eta}{2}\mathrm{Tr}[\tilde{A}e^{-2\lambda\tilde{A}}\Sigma(\theta^*)] + 2\gamma(\theta^*)^T e^{2\lambda\tilde{A}}\tilde{A}\theta^* \\
&= \sum_i -\frac{\eta}{2}\mu_i e^{-2\lambda\mu_i}(n_i^T\Sigma(\theta^*)n_i) + 2\gamma\mu_i e^{2\lambda\mu_i}(n_i^T\theta_i^*)^2,
\end{aligned}
\tag{31}
$$

where $\mu_i, n_i$ are eigenvalues of eigenvectors of the symmetric matrix $\tilde{A}$. Therefore,

$$
I'(\lambda) = \sum_i \mu_i^2(\eta e^{-2\lambda\mu_i}n_i^T\Sigma(\theta^*)n_i + 4\gamma e^{2\lambda\mu_i}(n_i^T\theta_i^*)^2) \geq 0.
\tag{32}
$$

We have $I(\lambda) \equiv 0$ (which happens only if $\tilde{A}$ is not full rank) or $I(\lambda)$ strictly monotonic. As $\theta^*$ is a local minimum, we have $I(0) = 0$, which gives (4). Then we have $I(\lambda) \equiv 0$ or $I(\lambda) = 0$ iff $\lambda = 0$, which finishes the proof. $\qquad\square$

## B.4  Proof of Theorem 2

*Proof.* By the definition of $K$-invariance, and taking derivative $\nabla_\theta$ of both sides of $\ell(x, K(\theta,\lambda)) = \ell(x,\theta)$, we have

$$
\nabla_\theta K(\theta,\lambda)^T \nabla_{K(\theta,\lambda)}\ell(x, K(\theta,\lambda)) = \nabla_\theta\ell(x,\theta),
\tag{33}
$$

where the l.h.s. follows from the chain rule. This imlies that

$$
(I + \lambda\nabla Q + O(\lambda^2))\nabla_{K(\theta,\lambda)}\ell(x, K(\theta,\lambda)) = \nabla_\theta\ell(x,\theta),
\tag{34}
$$

and so

$$
\nabla_{K(\theta,\lambda)}\ell(x, K(\theta,\lambda)) = (I - \lambda\nabla Q + O(\lambda^2))\nabla_\theta\ell(x,\theta),
\tag{35}
$$

If $F$ is $K$-invariant, the following Equation holds:

$$
\ell(x, K(\theta,\lambda)) + \|\nabla_{K(\theta,\lambda)}\ell(x, K(\theta,\lambda))\|^2 + \|K(\theta,\lambda)\|^2 = \ell(x,\theta) + \|\nabla_\theta\ell(x,\theta)\|^2 + \|\theta\|^2.
\tag{36}
$$

By the assumption, $\ell(x, K(\theta,\lambda)) = \ell(x,\theta)$, and (36), we have that

$$
\begin{aligned}
\|\nabla_\theta\ell(x,\theta)\|^2 + \|\theta\|^2 &= \|\nabla_{K(\theta,\lambda)}\ell(x, K(\theta,\lambda))\|^2 + \|K(\theta,\lambda)\|^2 \tag{37} \\
&= \|\nabla_\theta\ell(x,\theta)\|^2 + \|\theta\|^2 + 2\lambda(\nabla^T\ell\nabla Q^T\nabla\ell - \gamma Q^T\theta) + O(\lambda^2). \tag{38}
\end{aligned}
$$

Thus,

$$
\eta\nabla^T\ell\nabla Q^T\nabla\ell - \gamma Q^T\theta = 0.
\tag{39}
$$

There are two cases: (1) $\eta\nabla^T\ell\nabla Q^T\nabla\ell - \gamma Q^T\theta = 0$, and (2) $\eta\nabla^T\ell\nabla Q^T\nabla\ell - \gamma Q^T\theta \neq 0$.

For case (1), we are done. For case (2), the equation cannot hold for $\gamma + d\gamma$ because the first term is independent of $\gamma$. Thus, we can only have case (1).

This means that for any $\theta$

$$
\|K(\theta,\lambda)\|^2 = \|\theta\|^2 + O(\lambda^2),
\tag{40}
$$

which is only possible if $\|K(\theta,\lambda)\|^2 = \|\theta\|^2$. This completes the proof. $\qquad\square$

## B.5 Proof of Theorem 3

*Proof.* By definition we have $\|O\theta\|^2 = \|\theta\|^2$. Take derivative on both sides of $\ell(x, O\theta) = \ell(x, \theta)$, we have

$$O^T \nabla_{O\theta} \ell(x, O\theta) = \nabla_\theta \ell(x, \theta). \tag{41}$$

Thus we have

$$\nabla_{O\theta} \ell(x, O\theta) = O^{-T} \nabla_\theta \ell(x, \theta), \tag{42}$$

which gives $\|\nabla_{O\theta} \ell(x, O\theta)\|^2 = \|O^{-T} \nabla_\theta \ell(x, \theta)\|^2 = \|\nabla_\theta \ell(x, \theta)\|^2$. Combining the above results we have $F_{\eta,\gamma}(O\theta) = F_{\eta,\gamma}(\theta)$. $\qquad\square$

## B.6 Proof of Theorem 5

*Proof.* Rescaling symmetry implies that if we make the transform

$$W_i \to e^\lambda W_i, \; W_j \to e^{-\lambda} W_j, \tag{43}$$

$L$ does not change.

This corresponds to the choice of

$$A_{klm}^{\tilde{k}\tilde{l}\tilde{m}} = \begin{cases} 1 & k = \tilde{k} = i \\ -1 & k = \tilde{k} = j \\ 0 & \text{otherwise} \end{cases} \tag{44}$$

in Theorem 4, where the index $klm$ corresponds to the $m$−th element of the $l$-th unit of the $k$−th layer. Then we have

$$\eta(\mathbb{E}\mathrm{Tr}[g_i g_i^T - g_j g_j^T]) = 4\gamma(\mathrm{Tr}[W_i W_i^T - W_j W_j^T]). \tag{45}$$

This finishes the proof by setting $\gamma = 0$. $\qquad\square$

## B.7 Proof of Theorem 6

*Proof.* We can choose $A$ to be a rescaling matrix w.r.t. the $j$−th neuron of the $i$−th layer. Specifically, we choose

$$A_{klm}^{\tilde{k}\tilde{l}\tilde{m}} = \begin{cases} 1 & k = \tilde{k} = i, l = \tilde{l} = j \\ -1 & k = \tilde{k} = i+1, m = \tilde{m} = j \\ 0 & \text{otherwise} \end{cases} \tag{46}$$

in Theorem 4, which gives

$$\eta\mathbb{E}\mathrm{Tr}[g_{i,j,:}g_{i,j,:}^\top - g_{i+1,:,j}g_{i+1,:,j}^\top] = 4\gamma\mathrm{Tr}[w_{i,j,:}w_{i,j,:}^T - w_{i+1,:,j}w_{i+1,:,j}^T]. \tag{47}$$

$\qquad\square$

## B.8 Gradient Imbalance in Polynomial Networks

**Theorem 11.** *(Neuron Balance) For all local minima of Eq. (3) and any $i$, $j$,*

$$\eta\mathbb{E}\mathrm{Tr}[g_{i,j,:}g_{i,j,:}^\top - dg_{i+1,:,j}g_{i+1,:,j}^\top] = 4\gamma\mathrm{Tr}[w_{i,j,:}w_{i,j,:}^T - dw_{i+1,:,j}w_{i+1,:,j}^T]. \tag{48}$$

This means that unless $d = 1$, these networks will either have a noise or weight explosion problem. If $\gamma = 0$, the gradient fluctuation grows like $d^D$, exponential in depth $D$. When $d < 1$, later layers will have an exploding noise; when $d > 1$, earlier layers will have an exploding noise. When both $\eta$ and $\gamma \neq 0$, the sum of the noise and gradient norm will explode exponentially. In some sense, this implies that linear or sublinear types of activations are the only stable activations for deep neural networks.

*Proof.* We can still choose $A$ to be a rescaling matrix, but this time we should rescale the $i+1$−th layer more

$$w_{i,j,:} \to e^\lambda w_{i,j,:}, \; w_{i+1,:,j} \to e^{-d\lambda} w_{i+1,:,j}. \tag{49}$$

This corresponds to

$$A^{\tilde{k}\tilde{l}\tilde{m}}_{klm} = \begin{cases} 1 & k = \tilde{k} = i, l = \tilde{l} = j \\ -d & k = \tilde{k} = i+1, m = \tilde{m} = j \\ 0 & \text{otherwise} \end{cases} \tag{50}$$

in Theorem 4, which gives

$$\eta \mathbb{E}\text{Tr}[g_{i,j,:}g^{\top}_{i,j,:} - dg_{i+1,:,j}g^{\top}_{i+1,:,j}] = 4\gamma \text{Tr}[w_{i,j,:}w^{T}_{i,j,:} - dw_{i+1,:,j}w^{T}_{i+1,:,j}]. \tag{51}$$

This gives

$$\mathbb{E}\text{Tr}[g_{i,j,:}g^{\top}_{i,j,:}] = d\mathbb{E}\text{Tr}[g_{i+1,:,j}g^{\top}_{i+1,:,j}] \tag{52}$$

by choosing $\gamma = 0$. $\qquad \square$

## B.9 Proof of Theorem 7

*Proof.* The double rotation symmetry can be written as

$$U \to e^{\lambda A}U, \ W \to e^{-\lambda A}W, \tag{53}$$

where $A$ is an arbitrary matrix. We can thus choose the following generator

$$A^{\tilde{k}\tilde{l}\tilde{m}}_{klm} = \begin{cases} 1 & k = \tilde{k} = 1, \ l = i, \tilde{l} = j, \ \text{or } l = j, \tilde{l} = i \\ -1 & k = \tilde{k} = 2, \ l = i, \tilde{l} = j, \ \text{or } l = j, \tilde{l} = i \\ 0 & \text{otherwise} \end{cases} \tag{54}$$

in Theorem 4, where $k = 1$ corresponds to $W$ and $k = 2$ corresponds to $U$. This gives

$$\sum_k \eta[G_{W_{ki}}G_{W_{kj}} - G_{U_{ik}}G_{U_{jk}}] = \sum_k 4\gamma[W_{ki}W_{kj} - U_{ik}U_{jk}], \tag{55}$$

which finishes the proof. $\qquad \square$

## B.10 Proof of Lemma 1

*Proof.* By definition, we have

$$e^{\lambda A}\nabla^2_{e^{\lambda A}\theta}\ell(x, e^{\lambda A}\theta)e^{\lambda A} = \nabla^2 \ell(x, \theta), \tag{56}$$

and thus

$$T(e^{\lambda A}\theta) = \text{Tr}[e^{-2\lambda A}\mathbb{E}\nabla^2 \ell(x, \theta)]. \tag{57}$$

Let $A := \sum_i \mu_i n_i n_i^T$, and thus

$$T(e^{\lambda A}\theta) = \sum_i e^{-2\lambda \mu_i}(n_i^T \mathbb{E}\nabla^2 \ell(x, \theta)n_i). \tag{58}$$

As $A\mathbb{E}\nabla^2 \ell(x, \theta) \neq 0$, there exists $i$ such that $\mu_i \neq 0$ and $n_i^T \mathbb{E}\nabla^2 \ell(x, \theta)n_i \neq 0$. Therefore, we have $\lim_{\lambda \to +\infty} |T(e^{\lambda A}\theta)| = +\infty$ if $\mu_i < 0$, and $\lim_{\lambda \to -\infty} |T(e^{\lambda A}\theta)| = +\infty$ if $\mu_i > 0$. $\qquad \square$

## B.11 Proof of Theorem 8

We first prove the following theorem, which we will leverage to prove Theorem 8.

**Theorem 12.** *Let $V' = \sqrt{\Sigma_\epsilon}V\sqrt{\Sigma_x}$ such that $V' = \tilde{U}S'\tilde{V}$ is its SVD and rank$(V') = d$. Assume that every layer has more than $d$ hidden units. Then if $\gamma = 0$ and $\eta = 0^+$, at any global minimum of (3), we have*

$$\sqrt{\Sigma_\epsilon}M_1W_D = \tilde{U}\Sigma_D U^T_{D-1}, \ W_i = U_i\Sigma_i U^T_{i-1}, \ W_1M_2M_3\sqrt{\Sigma_x} = U_1\Sigma_1\tilde{V}, \tag{59}$$

*for $i = 2, \cdots, D-1$, where $U_i$ are arbitrary matrices satisfying $U_i^T U_i = I_{d \times d}$, and $\Sigma_x = \mathbb{E}[xx^T]$, $\Sigma_\epsilon = \mathbb{E}[\epsilon\epsilon^T]$. Moreover,*

$$\Sigma_1 = (\text{Tr}S')^{-\frac{D-2}{2D}}\frac{\text{Tr}[M_2M_3\Sigma_x M_3^T M_2^T]^{\frac{D-1}{2D}}}{\text{Tr}[M_1^T \Sigma_\epsilon M_1]^{\frac{1}{2D}}}\sqrt{S'},$$

$$\Sigma_D = (\text{Tr}S')^{-\frac{D-2}{2D}}\frac{\text{Tr}[M_1^T \Sigma_\epsilon M_1]^{\frac{D-1}{2D}}}{\text{Tr}[M_2M_3\Sigma_x M_3^T M_2^T]^{\frac{1}{2D}}}\sqrt{S'}, \tag{60}$$

$$\Sigma_i = (\text{Tr}S')^{1/D}(\text{Tr}[M_1^T \Sigma_\epsilon M_1]\text{Tr}[M_2M_3\Sigma_x M_3^T M_2^T])^{-\frac{1}{2D}}I_d.$$

*Proof.* Consider two consecutive layers $W_i$ and $W_{i+1}$. Using Theorem 7, we have

$$\eta \mathbb{E}[G_{W_{i+1}}^T G_{W_{i+1}} - G_{W_i} G_{W_i}^T] = 0. \tag{61}$$

By the MSE loss $\ell(x, y) = \|y - M_1 W_D \cdots W_1 M_2 M_3 x\|^2$, this gives

$$W_i h_i \mathbb{E}[\|\xi_{i+1}^T \tilde{r}\|^2 \tilde{x} \tilde{x}^T] h_i^T W_i^T = W_{i+1}^T \xi_{i+1}^T \mathbb{E}[\|h_i \tilde{x}\|^2 \tilde{r} \tilde{r}^T] \xi_{i+1} W_{i+1}, \tag{62}$$

where $\tilde{x} = \mathbb{E}_{x \in \mathcal{B}} x$ and $\tilde{r} := \mathbb{E}_{x \in \mathcal{B}}[y - M_1 W_D \cdots W_1 M_2 M_3 x]$ satisfy $\mathbb{E}\tilde{x} = \mathbb{E}\tilde{r} = 0$, and thus $\mathbb{E}\tilde{x}\tilde{x}^T = \frac{\Sigma_x}{|\mathcal{B}|}$ and $\mathbb{E}\tilde{\epsilon}\tilde{\epsilon}^T = \frac{\Sigma_\epsilon}{|\mathcal{B}|}$. We use the fact that at the global minimum we have $M_1 W_D \cdots W_1 M_2 M_3 = V$, and thus $y - M_1 W_D \cdots W_1 M_2 M_3 x$ is independent of $x$. We denote $\xi_{i+1} := M_1 W_D \cdots W_{i+2}$, $h_i := W_{i-1} \cdots W_1 M_2 M_3$ for $i = 2, 3, \cdots, D-2$, and $\xi_D := M_1$, $h_1 := M_2 M_3$.

Finally denote $W_1' = W_1 M_2 M_3 \sqrt{\Sigma_x}$ and $W_D' = \sqrt{\Sigma_\epsilon} M_1 W_D$, which gives $W_D' \cdots W_1' = V'$ and

$$W_{i+1}^\top \frac{W_{i+2}^\top \cdots W_D'^\top W_D' \cdots W_{i+2}}{\mathrm{Tr}\left[W_{i+2}^\top \cdots W_D'^\top W_D' \cdots W_{i+2}\right]} W_{i+1} = W_i \frac{W_{i-1} \cdots W_1' W_1'^\top \cdots W_{i-1}^\top}{\mathrm{Tr}\left[W_{i-1} \cdots W_1' W_1'^\top \cdots W_{i-1}^\top\right]} W_i^\top \tag{63}$$

for $i = 2, 3, \cdots, D-2$. For $i = 1$ we have

$$W_2^\top \frac{W_3^\top \cdots W_D'^\top W_D' \cdots W_3}{\mathrm{Tr}\left[W_3^\top \cdots W_D'^\top W_D' \cdots W_3\right]} W_2 = \frac{W_1' W_1'^\top}{\mathrm{Tr}\left[M_2 M_3 \Sigma_x M_2^\top M_3^\top\right]} \tag{64}$$

and for $i = D-1$ we have

$$\frac{W_D'^\top W_D'}{\mathrm{Tr}\left[M_1^T \Sigma_\epsilon M_1\right]} = W_{D-1} \frac{W_{D-2} \cdots W_1' W_1'^\top \cdots W_{D-2}^\top}{\mathrm{Tr}\left[W_{D-2} \cdots W_1' W_1'^\top \cdots W_{D-2}^\top\right]} W_{D-1}^\top. \tag{65}$$

Lemma 2 proves that we can decompose the matrices $W_1', W_2, \cdots, W_{D-1}, W_D'$ as

$$W_D' = U_D \Sigma_D U_{D-1}^T, \ W_{D-1} = U_{D-1} \Sigma_{D-1} U_{D-2}^T, \cdots, \ W_1' = U_1 \Sigma_1 U_0. \tag{66}$$

By minimizing (3) at $\eta = 0^+$, we need to minimize

$$\mathbb{E}\|y - W_D \cdots W_1 x\|^2 = \mathbb{E}\|\epsilon\|^2 + \|(V - W_D \cdots W_1)\sqrt{\Sigma_x}\|^2. \tag{67}$$

Thus we have

$$(V - W_D \cdots W_1)\sqrt{\Sigma_x} = 0, \tag{68}$$

which gives

$$W_D' \cdots W_1' = \sqrt{\Sigma_\epsilon} V \sqrt{\Sigma_x} = V'. \tag{69}$$

Then we obtain $U_D = \tilde{U}$, $U_0 = \tilde{V}$ and

$$\Sigma_D \Sigma_{D-1} \cdots \Sigma_1 = S'. \tag{70}$$

We can assume $\Sigma_D, \cdots, \Sigma_1 \in \mathbb{R}^{d \times d}$ because their ranks are the same by (63), (64) and (65).
(63) gives

$$\frac{\Sigma_{i+1}^2 \Sigma_{i+2}^2 \cdots \Sigma_D^2}{\mathrm{Tr}[\Sigma_{i+2}^2 \cdots \Sigma_D^2]} = \frac{\Sigma_1^2 \cdots \Sigma_{i-1}^2 \Sigma_i^2}{\mathrm{Tr}[\Sigma_1^2 \cdots \Sigma_{i-1}^2]}, \tag{71}$$

and thus $\Sigma_i = c I_d$ for $i = 2, 3, \cdots, D-2$ and

$$\frac{\Sigma_1^2}{\mathrm{Tr}[\Sigma_1^2]} = \frac{\Sigma_D^2}{\mathrm{Tr}[\Sigma_D^2]}. \tag{72}$$

(64) and (65) give

$$\frac{\Sigma_2^2 \Sigma_3^2 \cdots \Sigma_D^2}{\mathrm{Tr}[\Sigma_3^2 \cdots \Sigma_D^2]} = \frac{\Sigma_1^2}{\mathrm{Tr}[M_2 M_3 \Sigma_x M_3^T M_2^T]}, \quad \frac{\Sigma_D^2}{\mathrm{Tr}[M_1^T \Sigma_\epsilon M_1]} = \frac{\Sigma_1^2 \cdots \Sigma_{D-1}^2}{\mathrm{Tr}[\Sigma_1^2 \cdots \Sigma_{D-2}^2]}, \tag{73}$$

and thus

$$c^2 \frac{\Sigma_D^2}{\mathrm{Tr}[\Sigma_D^2]} = \frac{\Sigma_1^2}{\mathrm{Tr}[M_2 M_3 \Sigma_x M_3^T M_2^T]}, \quad \frac{\Sigma_D^2}{\mathrm{Tr}[M_1^T \Sigma_\epsilon M_1]} = c^2 \frac{\Sigma_1^2}{\mathrm{Tr}[\Sigma_1^2]}. \tag{74}$$

Combining (70), (72) and (74), we finish the proof.

$$\square$$

Now, we are ready to prove Theorem 8.

*Proof.* By Theorem 12, at any global minimum of (3), the solution of a $D_A$-layer network for the dataset $\mathcal{D}_M$ is given by

$$\sqrt{\Sigma_\epsilon} M_1 W_{D_A}^A = \tilde{U} \Sigma_D U_{D_A-1}^T, \ W_i^A = U_i \Sigma_i U_{i-1}^T, \ W_1^A M_2 M_3 \sqrt{\Sigma_x} = U_1 \Sigma_1 \tilde{V} \tag{75}$$

for $i = 2, \cdots, D-1$, where $U_i$ are arbitrary matrices satisfying $U_i^T U_i = I_{d \times d}$, and

$$\Sigma_1 \propto \sqrt{S'}, \ \Sigma_D \propto \sqrt{S'}, \ \Sigma_i \propto I_d \tag{76}$$

for some constants $c_1, c_2, c_3$. The solution suggests that

$$h_A^{L_A}(x) = \Pi_{i=1}^{L_A} W_i^A M_2 M_3 x \propto U_{L_A} \sqrt{S'} \tilde{V} \Sigma_x^{-1/2} x. \tag{77}$$

Similarly

$$h_B^{L_B}(x) = \Pi_{i=1}^{L_B} W_i^B M_2' M_3' x \propto U_{L_B} \sqrt{S'} \tilde{V} \Sigma_x^{-1/2} x. \tag{78}$$

The proof is complete by comparing (77) and (78). □

One might also consider the case where the minimal width of network B is $d_B < d$. In this case, we denote $\bar{S}' \in \mathbb{R}^{d_B \times d_B}$ containing top $d_B$ values of $S'$. Then we have

$$h_B^{L_B}(x) = \Pi_{i=1}^{L_B} W_i^B M_2' M_3' \propto U_{L_B} \sqrt{\bar{S}'} \tilde{V} \Sigma_x^{-1/2} x. \tag{79}$$

It is now not fully aligned with $h_A(x)$. To calculate the alignment, the corresponding kernels are

$$K_A(x_1, x_2) = h_A^{L_A}(x_1)^T h_A^{L_A}(x_2) = c_1 x_1^T \Sigma_x^{-1/2} \tilde{V}^T S' \tilde{V} \Sigma_x^{-1/2} x_2 \tag{80}$$

and

$$K_B(x_1, x_2) = h_B^{L_B}(x_1)^T h_B^{L_B}(x_2) = c_2 x_1^T \Sigma_x^{-1/2} \tilde{V}^T \bar{S}' \tilde{V} \Sigma_x^{-1/2} x_2 \tag{81}$$

for some constants $c_1, c_2 > 0$. We then have

$$\langle K_A, K_A \rangle_F = \mathbb{E} K_A(x_1, x_2)^2 = c_1^2 \mathrm{Tr}[(S')^2]. \tag{82}$$

Similarly we have

$$\langle K_B, K_B \rangle_F = c_2^2 \mathrm{Tr}[(\bar{S}')^2] \tag{83}$$

and

$$\langle K_A, K_B \rangle_F = \mathbb{E} K_A(x_1, x_2) K_B(x_1, x_2) = c_1 c_2 \mathrm{Tr}[\bar{S}' S']. \tag{84}$$

Therefore, the alignment is given by

$$\frac{\langle K_A, K_B \rangle_F}{\sqrt{\langle K_A, K_A \rangle_F \langle K_B, K_B \rangle_F}} = \frac{\mathrm{Tr}[\bar{S}' S']}{\sqrt{\mathrm{Tr}[(S')^2] \mathrm{Tr}[(\bar{S}')^2]}} = \sqrt{\frac{\mathrm{Tr}[(\bar{S}')^2]}{\mathrm{Tr}[(S')^2]}}, \tag{85}$$

which is some value between $0$ and $1$.

In the end of this section we proves the following technical lemma.

**Lemma 2.** *Suppose that matrices $W_1, W_2, \cdots, W_D$ satisfy*

$$W_{i+1}^T W_{i+2}^T ... W_D^T W_D ... W_{i+2} W_{i+1} = \lambda_i W_i W_{i-1} ... W_1 W_1^T ... W_{i-1}^T W_i \tag{86}$$

*for some $\lambda_i > 0$ and $i = 1, 2, \cdots, D-1$, then we have write the SVD of $W_1, W_2, \cdots, W_D$ as*

$$W_D = U_D \Sigma_D U_{D-1}^T, \ W_{D-1} = U_{D-1} \Sigma_{D-1} U_{D-2}^T, \cdots, \ W_1 = U_1 \Sigma_1 U_0, \tag{87}$$

*where $\Sigma_D, \cdots, \Sigma_1$ are the singular values.*

*Proof.* Denote $P_i := (W_i \cdots W_1)(W_i \cdots W_1)^T$. We then have

$$W_{i+1}^T W_{i+2}^T ... W_D^T W_D ... W_{i+2} W_{i+1} = \lambda_i P_i \tag{88}$$

and

$$W_{i+2}^T ... W_D^T W_D ... W_{i+2} = \lambda_{i+1} W_{i+1} P_i W_i^T, \tag{89}$$

which gives

$$P_i = \frac{\lambda_{i+1}}{\lambda_i} S_{i+1} P_i S_{i+1}, \tag{90}$$

where $S_{i+1} := W_{i+1}^T W_{i+1}$.

Suppose that $S_{i+1} = V\Lambda V^T$, and thus we have

$$A = c\Lambda A\Lambda, \tag{91}$$

where $c := \frac{\lambda_{i+1}}{\lambda_i}$ and $A := V^T P_i V$. The $(j, k)$ element gives

$$A_{jk}(1 - c\lambda_j\lambda_k) = 0. \tag{92}$$

If $A_{jk} \neq 0$, as $A$ is semi-definite, we also have $A_{jj}, A_{kk} \neq 0$, which gives $c\lambda_i^2 = c\lambda_j^2 = 1$. Therefore, we have $A\Lambda = \Lambda A$, and thus

$$P_i S_{i+1} = S_{i+1} P_i. \tag{93}$$

This shows that $P_i$ and $S_{i+1}$ share the same eigenspace. Moreover, we have $P_i = W_i P_{i-1} W_i^T$. Denote $W_i = U_i \Sigma_i U_{i-1}^T$ and $W_{i+1} = U_{i+1}\Sigma_{i+1}V$. As $P_{i-1}$ share the same eigenspace with $W_i^T W_i$, we have $P_{i-1} = U_{i-1}\Lambda U_{i-1}^T$ for some diagonal matrix $\Lambda$, which gives $P_i = U_i\Sigma_i\Lambda\Sigma_i U_i$. As $P_i$ share the same eigenspace with $W_{i+1}^T W_{i+1} = V^T\Sigma_{i+1}^2 V$, we obtain $V = U_i$, which finishes the proof. $\square$

### B.12 Proof of Theorem 9

*Proof.* By Theorem 7 we have

$$W_{i+1}^T W_{i+1} = W_i W_i^T \tag{94}$$

for $i = 1, \cdots, D - 1$, which suggests that $W_i = U_i\Sigma_i U_{i-1}^T$ with $\Sigma_i^2 = \Sigma_{i+1}^2$. At the global minimum we have $U_D(\Pi_{i=1}^D \Sigma_i)U_0^T = \Pi_{i=1}^D W_i = M_1^{-1} V M_2^{-1}$, which shows that the left side is the SVD of $M_1^{-1}V M_2^{-1}$. This finishes the proof. $\square$

### B.13 Proof of Theorem 10

*Proof.* By Theorem 12 we have

$$\sqrt{\Sigma_\epsilon}U = \frac{\text{Tr}[\Sigma_\epsilon]^{\frac{1}{4}}}{\text{Tr}[\Sigma_x]^{\frac{1}{4}}}\tilde{U}\sqrt{S'}U_1^T, \quad W\sqrt{\Sigma_x} = \frac{\text{Tr}[\Sigma_x]^{\frac{1}{4}}}{\text{Tr}[\Sigma_\epsilon]^{\frac{1}{4}}}U_1\sqrt{S'}\tilde{V}. \tag{95}$$

By [31, Proposition 5.3] we have

$$\begin{aligned} S(\theta) &= d_y\text{Tr}[W\Sigma_x W^T] + \|U\|_F^2\text{Tr}[\Sigma_x] \\ &= d_y\sqrt{\frac{\text{Tr}[\Sigma_x]}{\text{Tr}[\Sigma_\epsilon]}}\text{Tr}[S'] + \sqrt{\text{Tr}[\Sigma_x]\text{Tr}[\Sigma_\epsilon]}\text{Tr}[\Sigma_\epsilon^{-1}\tilde{U}S'\tilde{U}^T] \end{aligned} \tag{96}$$

This finishes the proof of (17).

Meanwhile we can also calculate

$$U^*, W^* = \arg\min_{U,W} d_y\|W\Sigma_x^{1/2}\|_F^2 + \|U\|_F^2\text{Tr}[\Sigma_x]. \tag{97}$$

As $UW\Sigma_x^{1/2} = V\Sigma_x^{1/2} := \hat{U}\hat{S}\hat{V}$, the minimum is given by

$$U^* = \left(\frac{\text{Tr}\Sigma_x}{d_y}\right)^{1/4}\hat{U}\sqrt{\hat{S}}\hat{U}_1, \quad W^*\Sigma_x^{1/2} = \left(\frac{\text{Tr}\Sigma_x}{d_y}\right)^{-1/4}\hat{U}_1\sqrt{\hat{S}}\hat{V} \tag{98}$$

and

$$\min S(\theta) = 2\sqrt{d_y\text{Tr}\Sigma_x}\text{Tr}\hat{S} \tag{99}$$

where $\hat{U}_1$ is an arbitrary orthogonal matrix. $\square$

