# OpenReview forum: "Neural Thermodynamics: Entropic Forces in Deep and Universal Representation Learning"
_NeurIPS.cc/2025/Conference — NeurIPS 2025 poster_

### Official Review · Reviewer_zJhT · 2025-07-01

**Clarity:** 3
**Significance:** 3
**Originality:** 3
**Rating:** 5
**Confidence:** 1

**Summary:**

This paper introduces a new theory for understanding the stochasticity and discretization effects arising from stochastic gradient descent (SGD). A key idea is approximating a single update of the original loss with n steps of updates of a loss, named entropic loss. When  $n \rightarrow \infty$, the discrete update is converted to a continuous update. The discretization error and noise effects are further decomposed from the entropic loss and termed effective entropy, and the gradient of the effective entropy, named entropic forces, is shown to break almost any continuous parameter symmetries while preserving discrete ones. As two example applications, the entropic forces are used to explain two observations: the progressive sharpening of the loss landscape and the emergence of universal representations

**Questions:**

- The entropic force appears to result from both stochasticity and discretization effects. Is it possible to analyze these two effects independently, or do they interact in a way that complicates their separation?
- When $n \rightarrow \infty$, the discrete update transitions to a continuous update for the effective entropy. Apart from the ability to use Lagrangian formalism, are there other practical advantages of this continuous update?
- In Figure 1, should the pink arrows be red instead?

**Ethical Concerns:**

["NO or VERY MINOR ethics concerns only"]

**Final Justification:**

My concerns are addressed and I will keep my original score

**Limitations:**

yes

**Paper Formatting Concerns:**

nil

**Quality:**

4

**Strengths And Weaknesses:**

Strengths And Weaknesses:

I have limited knowledge in this specific area and the overall manuscript appears satisfactory to me.

Quality:

- [+] The theoretical demonstrations are clear and seems technically sound.

Clarity:

- [+] The presentation of the idea is clear.

Significance& Originality:

- [+] I think this work is novel and can be a contribution to the Machine learning community.
- [+] New theory is introduced for understanding the learning dynamics.

---

> ### Author Rebuttal · Authors · 2025-07-29
>
> Thank you for your positive feedback and the detailed questions. We answer both the weaknesses and questions below.
>
> **Q1. The entropic force appears to result from both stochasticity and discretization effects. Is it possible to analyze these two effects independently, or do they interact in a way that complicates their separation?**
>
> Thank you for the insightful question. Yes, these two effects can be separated. The entropy term can be written as
> $$\eta\mathbb{E}_\mathcal{B}||\mathbb{E}\_{x\in\mathcal{B}}\nabla \ell(x,\theta)  ||^2=\eta[\mathbb{E}\_x\nabla \ell(x,\theta)]^2+\eta Var\_\mathcal{B}(\mathbb{E}\_{x\in\mathcal{B}}  \nabla \ell(x,\theta))$$
>
> The first term corresponds to the discretization effect with strength \eta, and the second term corresponds to the stochastic effect with strength $\eta/|\mathcal{B}|$. Thus, we can analyze each term separately.
>
> **Q2. When $n\to\infty$, the discrete update transitions to a continuous update for the effective entropy. Apart from the ability to use Lagrangian formalism, are there other practical advantages of this continuous update?**
>
> In our formalism, the continuous-time formalism serves primarily as a theoretical tool for understanding
> SGD training. For example, [16] uses it to argue that directly optimizing the effective loss via gradient descent does reproduce the same behavior as SGD. From a practical standpoint, however, SGD remains more effective and easier to implement than such continuous dynamics.
>
> **Q3. In Figure 1, should the pink arrows be red instead?**
>
> Thank you for catching that — yes, the pink arrows should be red. We will correct this in the revised version.

---

> > ### Author Response · Authors · 2025-08-04
> >
> > Dear reviewer,
> >
> > Thank you for your constructive feedback on our paper. Since we have not seen further updates,  we wanted to kindly check if our responses resolved the issues you raised. If you have any additional question, we are happy to clarify them and revise the manuscript accordingly.
> >
> > Thank you again for your valuable input!
> >
> > Best regards,
> > Authors

---

> > ### Comment · Reviewer_zJhT · 2025-08-05
> >
> > Thank the reviewer for addressing my concerns. I will keep my original score

---

### Official Review · Reviewer_oJQv · 2025-07-02

**Clarity:** 3
**Significance:** 3
**Originality:** 3
**Rating:** 5
**Confidence:** 2

**Summary:**

This paper introduces a theoretical framework that explains various emergent behaviors in deep learning by modeling stochasticity in the learning process caused by discrete time updates as governed by the entropic forces. The authors derived an entropic loss function for neural network learning and showed that this loss function breaks continuous symmetries while preserving discrete ones. The paper provides a theoretical explanation for various emergent phenomena in deep learning including gradient and weight alignment, universal feature structure, and sharpness seeking behaviors.

**Questions:**

1.	I’m not sure I understood Definition 1. First, the weight decay gamma does not appear in the definition. Second, do you mean that for a given function K satisfying locality and consistency, the loss function satisfies the invariance condition? In that case maybe switch the order of the 3 conditions.
2.	The entropic loss is initially denoted as phi but later denoted as F, make sure the notations are consistent or state explicitly otherwise.
3.	The gradient balancing result is interesting. However, I do not fully understand its connection to the results with small initialization, in your case, the weight balance requires eta goes to zero, so entropic force goes to zero and the system can be viewed as continuous gradient flow, but there’s no assumption on the initialization, so I’m wondering where this difference comes from.
4.	In Figure 2, the ranges of loss/entropy/imbalance are all relatively small, is this because you only select time points at later stage of training, i.e. close to equilibrium? Or is there some other reason why this is happening?
5.	In earlier relevant works on deep linear networks (i.e, Hanin & Zlokapa, Li & Sompolinsky), they also discussed and calculated the hidden representations, especially the average of the inner product of the hidden layer activation, across layers after training. Their calculation would be at the equilibrium of the loss, where no entropic loss is considered, and also averaged over all possible solutions, whereas your results are about learning with SGD and discrete dynamics without averaging over the solution space. While these two frameworks are very different, I’m wondering whether some results can be connected, i.e, can your results reduce to theirs in certain limits? Does the representation in your case also have the inner product aligning with the target output?
6.	In Figure 3, did you experiment with different types of nonlinearity to see how sensitive the alignment is? For example, how would alignment for a ReLU network look like?
7.	Some typo in Figure 4, I suppose these are mlp/attention layers and self/other alignment respectively but the axis labels are all the same.
8.	In section 4, does the balance result depend on the MSE loss? If not, why would you do a student-teacher setup with MSE loss on MNIST, it would seem more natural to simply train the original MSE dataset with CE loss?

**Ethical Concerns:**

["NO or VERY MINOR ethics concerns only"]

**Final Justification:**

My questions regarding connections to relevant works, clarifying the assumptions of the theorems and better justify the setups of some of the experiments have been resolved by the rebuttal. I keep my score unchanged, this paper provides valuable insights in understanding the learning dynamics of neural networks in discrete time and is a valuable addition to the field of theoretical ml.

**Limitations:**

Yes.

**Paper Formatting Concerns:**

No major formatting issues.

**Quality:**

3

**Strengths And Weaknesses:**

Strengths: The perspective of using an entropic loss to understand stochasticity in the learning dynamics is quite interesting and novel. The paper then proceeds with rigorous theoretical proofs to demonstrate interesting implications of this perspective. Using this entropic loss, the paper shows that the loss breaks all continuous symmetry and preserves discrete ones, this is in stark contrast with previous works that focus on the equilibrium of continuous gradient flow without incorporating the stochasticity induced by discrete time dynamics. The framework also provides explanations for previously observed behaviors including weight and gradient alignment, universal feature structure and sharpness seeking. All results are supported with sufficient numerical experiments.
This paper not only provides a new insights on understanding learning dynamics in discrete time with mathematical rigor, but also discusses thoroughly the potential implications of the theoretical results and their connections to widely observed phenomenon. Therefore, I believe this paper is a valuable addition to theoretical understanding of ML and is of interest to a broad audience.

Weaknesses:
While the paper shows empirical evidence in realistic settings. Many of the main theorems (8-10) are established for deep linear networks, or particular types of settings (theorems 5-6) I think it is still insightful as long as the theoretical results can predict behaviors in more complicated settings, as the authors show, but it would be nice to have a summary of results or discussions which part of the results are applied to which types of networks/loss etc.

---

> ### Author Rebuttal · Authors · 2025-07-29
>
> Thank you for your positive feedback and the detailed questions. We answer both the weaknesses and questions below.
>
> **Q1.While the paper shows empirical evidence in realistic settings. Many of the main theorems (8-10) are established for deep linear networks, or particular types of settings (theorems 5-6) I think it is still insightful as long as the theoretical results can predict behaviors in more complicated settings, as the authors show, but it would be nice to have a summary of results or discussions which part of the results are applied to which types of networks/loss etc.**
>
> Thank you for the suggestion. Having a table to summarize the applicability of these results would be great. Theorems 1–4 are general and do not depend on specific network architectures or loss functions. Theorems 5–6 apply to arbitrary ReLU layers, Theorem 7 applies to linear or self-attention layers, and Theorems 8–10 concern deep linear networks trained with MSE loss. We will add a summary table or paragraph to clarify these conditions in the main text.
>
> **Q2. I’m not sure I understood Definition 1. First, the weight decay gamma does not appear in the definition. Second, do you mean that for a given function K satisfying locality and consistency, the loss function satisfies the invariance condition? In that case maybe switch the order of the 3 conditions.**
>
> We appreciate your thoughtful comments. You are right on both points. The weight decay parameter $\gamma$ was inadvertently included in Definition 1 and will be removed. Indeed, the invariance condition of the loss function holds when the function K satisfies both locality and consistency. As suggested, we will reorder the conditions to first state locality and consistency for K, then the invariance property. This restructuring will better reflect the logical flow of our definition. Thank you for catching these important points.
>
>
> **Q3. The entropic loss is initially denoted as phi but later denoted as F, make sure the notations are consistent or state explicitly otherwise.**
>
> Thank you for catching this. The entropic loss F is defined as the expectation of $\phi$ (lines 97–98), so they are related but not identical.
>
> **Q4.The gradient balancing result is interesting. However, I do not fully understand its connection to the results with small initialization, in your case, the weight balance requires eta goes to zero, so entropic force goes to zero and the system can be viewed as continuous gradient flow, but there’s no assumption on the initialization, so I’m wondering where this difference comes from.**
>
> This is an insightful observation. The key point is that Theorem 4 does not rely on any assumptions about initialization or learning rate. It describes the structure of a local minimum of equation (3), arising purely from symmetry and weight decay — independent of optimization dynamics or initialization scale.
>
> While the entropic force vanishes as $\eta \to 0$, the weight balance induced by symmetry remains nontrivial, as discussed in lines 155–158. This is because the weight balance is due to weight decay, and already encourages the weights to be small (thus, sometimes leading to a similar effect to a small init.).
>
> **Q5.In Figure 2, the ranges of loss/entropy/imbalance are all relatively small, is this because you only select time points at later stage of training, i.e. close to equilibrium? Or is there some other reason why this is happening?**
>
> The small ranges in loss, entropy, and imbalance are task-specific. In fact, the absolute values of them are not quite meaningful. For this task, we have a small initial imbalance (blue points). During training, entropy and imbalance decrease further. The plot includes the entire training trajectory, not just the equilibrium phase.
>
> **Q6.In earlier relevant works on deep linear networks (i.e, Hanin & Zlokapa, Li & Sompolinsky), they also discussed and calculated the hidden representations, especially the average of the inner product of the hidden layer activation, across layers after training. Their calculation would be at the equilibrium of the loss, where no entropic loss is considered, and also averaged over all possible solutions, whereas your results are about learning with SGD and discrete dynamics without averaging over the solution space. While these two frameworks are very different, I’m wondering whether some results can be connected, i.e, can your results reduce to theirs in certain limits? Does the representation in your case also have the inner product aligning with the target output?**
>
> Indeed, our settings are quite different. Both Hanin & Zlokapa, Li & Sompolinsky consider the Bayesian setting, where the noise is thermal and equivalently generated by an isotropic Gaussian process. Our work directly deals with the actual SGD noise coming from the minibatch sampling. Many predictions of the two theories are fundamentally and qualitatively different, for example, the Bayesian type theory (such as Hanin & Zlokapa, Li & Sompolinsky) predicts a nonvanishing stationary probability measure everywhere, but the actual trajectories of SGD typically occupy a small subset of the entire space, because the entropic contribution is not negligible even at small \eta and \gamma.
>
> There are a few aspects that the two types of theories do agree on. For example, as shown in the proof of Theorem 8, the learned representation h(x) satisfies $h(x)^T h(x) \propto (Vx)^T (Vx)$, indicating alignment with the target output. Exploring deeper connections with prior equilibrium analyses is an interesting future direction.
>
>
> **Q7.In Figure 3, did you experiment with different types of nonlinearity to see how sensitive the alignment is? For example, how would alignment for a ReLU network look like?**
>
> Yes, we experimented with ReLU networks and found that the alignment pattern is nearly identical to that shown in the left panel of Figure 3. We used tanh purely for illustration, and will clarify this in the caption.
>
> **Q8.Some typo in Figure 4, I suppose these are mlp/attention layers and self/other alignment respectively but the axis labels are all the same.**
>
> Thank you for pointing this out. The first and third panels in Figure 4 are self-attention layers. We will correct the labels accordingly in the revision.
>
> **Q9.In section 4, does the balance result depend on the MSE loss? If not, why would you do a student-teacher setup with MSE loss on MNIST, it would seem more natural to simply train the original MSE dataset with CE loss?**
>
> Thanks for the comment. Both settings are fine, but we wanted to be consistent with other experiments and chose to have a more consistent setting with other experiments. We will also include the more natural setting in the final revision.

---

> > ### Author Response · Authors · 2025-08-04
> >
> > Dear reviewer,
> >
> > Thank you for your constructive feedback on our paper. Since we have not seen further updates,  we wanted to kindly check if our responses resolved the issues you raised. If you have any additional question, we are happy to clarify them and revise the manuscript accordingly.
> >
> > Thank you again for your valuable input!
> >
> > Best regards,
> > Authors

---

> > ### Comment · Reviewer_oJQv · 2025-08-05
> >
> > Thank you for the detailed response to my review. I keep my score unchanged and believe that the paper should be accepted.

---

### Official Review · Reviewer_UAQt · 2025-07-03

**Clarity:** 3
**Significance:** 2
**Originality:** 3
**Rating:** 5
**Confidence:** 2

**Summary:**

This paper tries to bridge the gap between theory and practice in deep learning. It studies a notion of entropic loss landscape and proves that the associated forces break continuous symmetries while preserving discrete ones, and gives a tradeoff between gradient and weight balancing. This also helps explain recent observed phenomena such as progressive sharpening and flattening in deep learning. Several experiments are run to support the claims.

**Questions:**

- What if the conditions on $\phi_{\eta}$ do not hold in Theorem 1 ? How often do they hold?
- What if the learning rate is not a matrix ? You said Adam corresponds to a matrix, what about other optimizers ?
- In Figure 2, is what you call entropy the $S(\theta)$ ?
- Does $\phi_{\eta}$ contain $\gamma$ in its definition? If so, how? If not, how did the gamma appear when computing its expectation?
- Can you say what the symmetries present (continuous and discrete) are in shallow linear network square loss for example ?
- Are there some losses in practice that have A-exponential symmetry ?
- How is theorem 2 saying that the continuous symmetry is broken ?
- Does the sharpness result change if we choose the largest singular value of the hessian instead of the trace?

**Ethical Concerns:**

["NO or VERY MINOR ethics concerns only"]

**Final Justification:**

My concerns were addressed

**Limitations:**

yes

**Paper Formatting Concerns:**

yes

**Quality:**

3

**Strengths And Weaknesses:**

Strengths:
- The research direction is interesting and may help explain some phenomena observed in practice in deep learning
- Results are supported with experiments.

Weaknesses:
- The results rely on strong assumptions.

---

> ### Author Rebuttal · Authors · 2025-07-28
>
> Thank you for your positive feedback and the detailed questions. We answer both the weaknesses and questions below. If you have additional questions, please raise them and we will be happy to address them.
>
> **Q1. The results rely on strong assumptions.**
>
> Thank you for your comment. Our theoretical framework (Theorems 4-7) is derived primarily from the existence of exponential symmetries, which are intrinsic properties of modern neural architectures, and they are not assumptions. For instance: rescaling symmetries of ReLU layers and rotational symmetry of attention layers. Also, see our answer below to Q6.
> These symmetries are structural characteristics of the architectures themselves, not additional assumptions imposed by our analysis. Consequently, our results reveal fundamental and **universal** properties that hold whenever these widely present symmetries exist, without requiring further restrictive conditions.
>
> **Q2. What if the conditions on \phi_\eta do not hold in Theorem 1 ? How often do they hold?**
>
> We thank the reviewer for raising this important question, which we will clarify. Regarding Theorem 1: The function $\phi_\eta$ is explicitly defined in L94-95 as an "equivalent" loss capturing the leading-order effect of gradient descent dynamics. This is not an imposed condition but a mathematical construction derived from Taylor expansion. The only technical requirement is the boundedness of the third derivative, which is a standard smoothness condition.
>
> **Q3. What if the learning rate is not a matrix ? You said Adam corresponds to a matrix, what about other optimizers ?**
>
> First of all, almost all commonly used gradient-based optimizers can be interpreted as gradient descent with a matrix-valued learning rate (SGD is an identity matrix learning rate, natural gradient descent has the inverse Fisher Information as the learning rate matrix, Newton’s method has the inverse Hessian as the learning rate, SignSGD has the inverse gradient as the learning rate). Some exceptions do exist — such as coordinate descent — but those methods are less relevant for deep learning and therefore fall outside the scope of our analysis.
>
> **Q4. In Figure 2, is what you call entropy the $S(\theta)$?**
>
> Yes. Throughout our paper, the entropy $S(\theta)$ refers to the third term in equation (3). For ReLU networks, such as those in Figure 2, $S(\theta)$ takes the simplified form given in equation (6).
>
> **Q5. Does $\phi_\eta$ contain \gamma in its definition? If so, how? If not, how did the gamma appear when computing its expectation?**
>
> As defined in eq.(3), the entropy $S(\theta)$ is independent of the weight decay $\gamma$ (a technical note: if $\gamma$ is very large, it can still appear in $S$ as a higher order term, but throughout our current work, $\gamma$ is regarded as a small quantity that its effect on $S$ can be ignored.). The learning rate $\eta$ and the weight decay $\gamma$ determine the balance between the gradients and the weights in Theorem 4.
>
> **Q6. Can you say what the symmetries present (continuous and discrete) are in shallow linear network square loss for example ?**
>
> Linear networks exhibit a lot of symmetries, the most common of which is what we referred to as the double rotational symmetry, meaning that two consecutive layers can be jointly transformed by a invertible matrix without changing the network output. When the network output itself is invariant to these changes, it also implies that the loss function is invariant. Therefore, we will only discuss why and how the network output is invariant.
>
> Consider a two-layer linear network $f(U,W,x)=UWx$. We have $f(UQ,Q^{-1}W,x)= UQQ^{-1}Wx = UWx = f(U,W,x)$, for any invertible matrix Q. This is what we call the double rotational symmetry, and is what we used to derive our gradient alignment result (Theorem 7). In fact, the double rotational symmetry includes both continuous and discrete symmetries (as subgroups) in linear networks.
>
> For example, the rescaling symmetry is a continuous symmetry: $f(\lambda U,\lambda^{-1}W,x)=f(U,W,x)$.
> Permutation symmetries are discrete symmetries: $(f(UP,P^{-1}W,x)=f(U,W,x)$ for any permutation matrix P. Of course, there is also the simplest type of discrete symmetry, the sign-flip symmetry (or, the $\mathbb{Z}_2$ symmetry): $f(U,W,x)= f(-U,-W,x)$. Note that all these symmetries are present for any $x$. Also, note that the loss function is invariant to these transformations for the same reason: $||f(U,W,x) - y||^2 =||f(-U,-W,x) - y||^2 $.
>
> Lastly, note that for a deeper network, different layers can have independent symmetries, and one can study them in isolation. For example, consider a three-layer linear network: $f(x) = UVW x$. There is a double rotation symmetry between $U$ and  $V$, and there is another double rotation symmetry between $V$ and $W$; our results can be applied to these two symmetries independently.
>
> **Q7. Are there some losses in practice that have A-exponential symmetry ?**
>
> Yes, there exist many practical losses exhibiting A-exponential symmetry, particularly in common neural network architectures. This should already become quite clear given our answer to Q6 above, but we will name a few more examples. Our analysis in Section 4 identifies two important cases:
>
> Rescaling Symmetry: In ReLU networks (Theorems 5-6), the function $f(U,W,x)=UReLU(Wx)$ satisfies $f(\lambda U,\lambda^{-1}W,x)=f(U,W,x)$ for any scalar $\lambda\neq0$. This symmetry exists in any architecture containing at least one ReLU layer, such as $f_1(UReLU(Wf_2(x)))$ where $f_1,f_2$ are arbitrary sub-networks.
>
> Double Rotational Symmetry: As discussed in Q6, linear and self-attention layers exhibit this property between the key and query matrices (Theorem 7), where the loss remains invariant under simultaneous orthogonal transformations of certain parameter groups.
>
> These represent concrete instances of A-exponential symmetry in practice, with the specific form depending on the network architecture's structure and activation functions, and independent of the chosen loss. Thus, any network containing at least one self-attention layer has one double rotation symmetry: If a network has 10 self-attention layers, then it also has ten independent double rotation symmetries.
>
> **Q8. How is theorem 2 saying that the continuous symmetry is broken ?**
>
> Theorem 2 states that if the loss $\ell$ is K-invariant but K does not satisfy conditions (1) or (2), then the associated $F$ is not K-invariant. In other words, the symmetry must disappear unless K corresponds to rotational invariance.
>
> **Q9. Does the sharpness result change if we choose the largest singular value of the hessian instead of the trace?**
>
> No, the main conclusion remains the same. We used the trace of the Hessian for analytical convenience. For example, Lemma 1 also implies that the largest singular value also diverges under the stated assumptions (because divergence of the trace of a PSD matrix happens if and only if its largest singular value diverges).

---

> > ### Author Response · Authors · 2025-08-04
> >
> > Dear reviewer,
> >
> > Thank you for your constructive feedback on our paper. Since we have not seen further updates,  we wanted to kindly check if our responses resolved the issues you raised. If you have any additional question, we are happy to clarify them and revise the manuscript accordingly.
> >
> > Thank you again for your valuable input!
> >
> > Best regards,
> > Authors

---

> > ### Comment · Reviewer_UAQt · 2025-08-07
> >
> > I thank the authors for their answer. I still don't understand Q5. I don't see any $\gamma$ in the definition of $\phi_{\eta}$ but after taking the expectation the $\gamma$ appears. Can you clarify this?

---

> > > ### Author Response · Authors · 2025-08-07
> > >
> > > Thank you for your question.
> > >
> > > To deal with the weight decay, technically, one should replace $\ell(x,\theta)$ with $\ell_{\gamma}(x,\theta):=\ell(x,\theta)+\gamma||\theta||^2$. Plug this definition of into Theorem 1, we would obtain the following $\gamma$-dependent term in the expression of $\phi$: $\gamma||\theta||^2+\gamma\theta^T\Lambda\theta+O(||\Lambda||^2\gamma)$, and we drop the second and third terms because in our current theory, we treat both $\eta$ and $\gamma$ as amll quantities. These terms contribute to $F$ at the strength of $\eta \gamma$, which is a higher-order small quantity, and are thus negligible in our theory. Then one can obtain eq.(3). We will clarify this technical point in the footnote.

---

> > > > ### Comment · Reviewer_UAQt · 2025-08-08
> > > >
> > > > I thank the authors for their clarification. My concerns were mainly addressed. I will increase my score.

---

### Note · Authors · 2025-08-14

Dear AC and Reviewers,

We sincerely thank the reviewers and AC for their insightful feedback. We are encouraged by the consensus that our entropic-force theory offers a crucial perspective on neural network dynamics.  It is also encouraging to see that all reviewers agree that our theory is validated by our experiments. In our final revision, we will certainly incorporate the comments of all reviewers and improve our manuscript therewith.

Here, we would like to emphasize and single out the potential broader implications of our work, which we believe are plenty and worthwhile to point out:

1. Our result is the first to give a direct proof of the Platonic representation hypothesis (PRH) in any scenario, and may become the cornerstone of developing more complicated theories in the future;

2. Our result offers a crucial link between the past works on the theory of SGD training (Smith et al., 2021) with important problems such as the PRH and progressive sharpening (PS);

3. Our result implies that many seemingly disconnected phenomena in deep learning may have a common cause, just like the PRH and PS; this may eventually lead to a bigger unified theory of the phenomenology of deep learning;

4. Our result offers a crucial link between theoretical physics and deep learning, because the entropic forces we study are directly due to the irreversibility of the SGD training dynamics, which has the same origin as the second law of thermodynamics and phenomena such as the arrow of time and the expansion of the universe (e.g., see DOI:10.1086/151073).

5. Irreversible processes (such as critical periods) also exist extensively in neuroscience, and our result may thus become a first step towards a unified theory of biological and artificial learning (e.g., doi: 10.1016/j.tins.2020.01.002).


In short, our paper proposes that the entropic forces and irreversible processes could serve as a theoretical cornerstone for understanding AI, unifying several seemingly unrelated phenomena, and potentially providing a profound connection to natural sciences.


Best regards,

The Authors

---

### Decision · Program_Chairs · 2025-09-17

**Decision:**

Accept (poster)

**Comment:**

This paper develops a framework for understanding learning dynamics in deep networks, with connections to symmetry breaking in representation learning, loss landscape sharpness, and emergence of universal representations.  After discussion with authors, all reviewers lean toward accept.  There is a consensus that the paper is technically solid and includes sufficient experimental evidence in real settings.  However, theorems involve strong assumptions and are for simpler settings or toy models (e.g., deep linear networks).  The Area Chair agrees with the reviewer consensus that the overall contribution justifies recommending acceptance.